# The Joint Effect of Mid-latitude Winds and the Westerly Quasi-Biennial Oscillation Phase on the Antarctic Stratospheric Polar Vortex and Ozone

Zhe Wang[1], Jiankai Zhang[1*], Siyi Zhao[1], Douwang Li[1]

[1]Key Laboratory for Semi-Arid Climate Change of the Ministry of Education, School of Atmospheric Sciences, Lanzhou University, Lanzhou, 730000, China.

*Correspondence to*: Jiankai Zhang (jkzhang@lzu.edu.cn)

**Abstract.** The quasi-biennial oscillation (QBO) dynamically interacts with the extratropical atmosphere. However, the relationship between the QBO in austral winter and the Antarctic stratospheric polar vortex in spring remains unclear. In this study, we propose a joint predictor involving the QBO for the Antarctic stratospheric polar vortex and ozone in austral spring. During the westerly QBO phase (WQBO), positive zonal-mean zonal wind anomalies at 20°S−40°S in the upper stratosphere in July, named as the positive extratropical mode, can lead to a stronger Antarctic stratospheric polar vortex and lower ozone concentration in November, with correlations reaching 0.75 and 0.60, respectively. The mechanism is summarized as follows: the positive extratropical mode triggers a secondary circulation, which further alters the environmental conditions for wave propagation in the stratosphere. The resulting anomalous wave divergence leads to a stronger Antarctic stratospheric polar vortex during the austral spring. While during the easterly QBO phase (EQBO), the correlation between the extratropical mode and the strength of the polar vortex is only 0.1. Due to the stronger upward motion in the tropics, which opposes the secondary circulation induced by the extratropical mode, the EQBO cannot sustain the positive anomalous zonal-mean zonal wind until November. Our results highlight that the extratropical mode during the WQBO could serve as a reliable predictor for both the Antarctic stratospheric polar vortex and the Antarctic ozone hole with a five-month time lag.

## 1 Introduction

The quasi-biennial oscillation (QBO) is a dominant mode of interannual variability in the tropical stratosphere (Lindzen and Holton 1968; Andrews and McIntyre 1976; Baldwin et al. 2001; Anstey and Shepherd, 2014; Yamashita et al., 2018; Rao et al., 2019; 2020a; 2023a). It is a quasi-periodic oscillation of the equatorial zonal wind between easterlies and westerlies in the tropical stratosphere, with a mean period of approximately 28 months. (Lindzen and Holton, 1968; Andrews and McIntyre, 1976; Baldwin et al., 2001). The QBO plays a crucial role in influencing atmospheric circulation and chemical species outside the tropical stratosphere (Holton and Tan, 1980; Ruti et al., 2006; Garfinkel and Hartmann, 2011; Yamashita et al., 2015; Gray et al., 2018; Rao et al., 2019; 2020a, b, c; Zhang et al., 2021). Holton and Tan (1980) proposed that the QBO can modify upward-propagating planetary waves by altering the zero-wind line in the stratosphere, which further affects the extratropical waveguide (Baldwin et al., 2001; Anstey et al., 2014; Zhang et al., 2019). During the westerly QBO phase (WQBO), the zero-wind line of the zonal-mean zonal wind shifts equatorward. This causes planetary waves to be reflected away from high

latitudes, leading to a strengthened stratospheric polar vortex by reducing wave-driven disturbances in the polar regions. Consequently, the Arctic stratospheric polar vortex tends to be stronger on average during the WQBO compared to the easterly QBO phase (EQBO). Additionally, Garfinkel et al. (2012) proposed that the secondary meridional circulation induced by the

EQBO also plays a crucial role in the Arctic stratosphere. This secondary circulation restricts the propagation of subpolar Rossby waves into the subtropics, resulting in more waves breaking closer to the pole. These QBO-induced changes in the Arctic stratospheric polar vortex can further influence the distribution of Arctic ozone and water vapor (Wang et al., 2022; Lu et al., 2023). Zhang et al. (2021) revealed that dynamical processes play a more important role in the QBO-Arctic ozone connection than chemical processes, although the impact of chemical processes is not negligible. Furthermore, observational

and modeling evidences both show that the QBO's influence on the Arctic stratospheric polar vertex and ozone occurs within the extended winter (November–March; Rao et al., 2019; 2020b; Zhang et al., 2021).

In the Southern Hemisphere (SH), upward-propagating planetary waves are weak due to the less topography and the weaker thermal contrast between land and sea. Consequently, the QBO-vortex coupling, which is closely related to the planetary waves, has received less attention than that in the Northern Hemisphere (NH, Garcia and Solomon, 1987; Lait et al.,

1989; Baldwin and Dunkerton, 1998; Naito, 2002; Hitchman and Huesmann, 2009; Yamashita et al. 2018; Rao et al., 2023a and b). Naito et al. (2002) examined the QBO signal in the SH and found that from September to October, the zonal-mean zonal wind in the lower stratosphere decelerates more rapidly during the EQBO than the WQBO. This deceleration is attributed to stronger upward wave propagation from the troposphere and larger wave convergence during the EQBO in these two months. However, Anstey et al. (2014) demonstrated that the extratropical response to the QBO occurs in November, interpreted as

modulation of the Antarctic stratospheric polar vortex's final warming. Thus, it remains unclear when the response of the Antarctic stratospheric polar vortex to the QBO reaches its peak. Yamashita et al. (2018) examined the influence of the QBO on SH extratropical circulation from austral winter to early summer using a multiple linear regression approach. Their findings suggest that the QBO-SH stratospheric polar vortex connection operates through two distinct pathways: the mid-stratospheric pathway, which tends to suppress the propagation of planetary waves into the stratosphere during the WQBO, and the low-

stratospheric pathway, which tends to enhance upward planetary waves in the EQBO. These QBO-SH stratospheric polar vortex connections established by Yamashita et al. (2018) are statistically significant. However, in QBO-resolving models from phases 5/6 of the Coupled Model Intercomparison Project (CMIP5/6), fewer than half of the General Circulation Models (GCM) successfully reproduce a weakened SH polar vortex during the EQBO (Rao et al., 2023b). Furthermore, they also suggested that even the high-skill models capture only about 30% of the observed deceleration in westerlies during the EQBO.

Although previous studies have revealed the potential relationships and mechanisms linking the QBO with the Antarctic stratospheric polar vortex, the weak correlation between them and the limited performance of GCMs indicate that the QBO-vortex coupling in the SH is not yet fully understood. In addition to the Antarctic stratospheric polar vortex, previous studies have also investigated the QBO signal in stratospheric ozone. Garcia and Solomon (1897) demonstrated that year-to-year fluctuations in Antarctic ozone may be linked to the tropical QBO. In contrast, Wang et al. (2022) suggested that the Antarctic

total column ozone (TCO) shows no significant response to the QBO signal (Figure 2 in Wang et al., 2022). So far, the impacts of the QBO on the Antarctic stratospheric polar vortex and ozone have not been well documented.

The QBO period varies irregularly between 17 and 38 months, which is considered as a reliable predictor of the stratospheric polar vortex, and further the near-surface weather and climate in the NH (Baldwin and Dunkerton, 2001; Zhang et al., 2020; Tian et al., 2023). In the SH, if QBO-vortex coupling does exist, it could also serve as a predictor of the Antarctic
stratospheric polar vortex and ozone. In this study, we aim to explore when and how the QBO influences the SH stratospheric polar vortex. Understanding the connection between the QBO and high-latitude circulation may help improve the forecasting accuracy of the SH stratospheric polar vortex, and, by extension, the Antarctic ozone hole. Section 2 presents the data and methods used in this research. In Section 3, we propose an improved predictor of the Antarctic stratospheric polar vortex and ozone based on the QBO. We also discuss the underlying mechanisms. The conclusions are provided in Section 4.

**2 Data and methods**

**2.1 Data**

Monthly and daily meteorological data and ozone volume mixing ratios are derived from Modern-Era Retrospective analysis for Research and Applications, Version 2 (MERRA-2; GMAO, 2015) reanalysis for the period from 1980 to 2022. The reanalysis data has a resolution of $1.25°\times1.25°$ and 42 pressure levels extending from 1000 hPa to 0.1 hPa. The Polar
Stratospheric Cloud (PSC) area is obtained from the NASA ozone watch website.

Previous studies have employed different methods to define the QBO index. Some are based on the tropical zonal-mean zonal wind at a single pressure level (Holton and Tan, 1980; Gray et al., 1992; Baldwin et al., 2001; Garfinkel and Hartmann, 2007), while others use two QBO indices at different pressure levels (Andrews et al., 2019) or apply empirical orthogonal function (EOF) analysis to the tropical zonal-mean zonal wind (Randel et al., 1999; Anstey et al., 2010; Rao and Ren, 2018).
The QBO phase defined by the EOF method is similar to that defined by the single pressure level (Baldwin et al., 2001; Rao et al., 2020b). Typically, when tropical stratospheric winds near 50 hPa are used to define the QBO phase, the NH stratospheric polar vortex during winter tends to strengthen during the WQBO and weaken during the EQBO (Baldwin et al. 2001; Anstey and Shepherd, 2014; Anstey et al., 2022). In the SH, the Antarctic stratospheric polar vortex shows responses to the tropical winds around 20 hPa (Baldwin and Dunkerton 1998; Baldwin et al., 2001; Baldwin and Dunkerton 1998; Anstey et al., 2022;
Rao et al., 2023b). Therefore, this study uses the standardized zonal-mean zonal wind averaged over 10°S−10°N at 20 hPa to define the QBO phase. The EQBO phase is defined as years when the tropical standardized zonal-mean zonal winds are less than −1, while the WQBO corresponds to years when the tropical standardized zonal-mean zonal winds exceed 1. The El Nino-Southern Oscillation (ENSO) index is defined as the sea surface temperature (SST) anomalies averaged over the Niño 3.4 region (5°N−5°S; 170°−120°W). SST data is derived from the monthly NOAA Extended Reconstruction SSTs Version 5
(ERSSTv5).

## 2.2 Method

### 2.2.1 Singular value decomposition (SVD)

The SVD analysis is performed between the zonal-mean zonal winds at latitudes ranging from 0° to 40°S and 1−70 hPa in July (extratropical mode) and the zonal-mean zonal winds at latitudes ranging from 50°S to 70°S and 1−70 hPa in November (polar mode). To validate the robustness of the findings, we conduct a Monte Carlo test (Iwasaka and Wallace, 1995) by creating 1000 ensembles of the SVD analysis. In this test, the extratropical wind field is kept fixed, while the polar wind field is randomly disordered in both time and space. The total variances of the 1000 paired ensembles are calculated and sorted. If the observed SVD total variance exceeds the 95[th] percentile of the ensemble distribution, the SVD mode is statistically significant at the 95% confidence level.

### 2.2.2 Eliassen-Palm (E-P) flux and mass stream function

The E-P flux (Andrews et al., 1987) is used to diagnose the propagation of waves, which is calculated as follows:

$$F_\varphi \equiv \rho_0 a cos\varphi \left( \frac{\overline{u}_z \overline{v'\theta'}}{\overline{\theta}_z} - \overline{u'v'} \right) \tag{1}$$

$$F_z \equiv \rho_0 a cos\varphi \left\{ \left[ f - (a cos\varphi)^{-1} (\overline{u} cos\varphi)_\varphi \right] \frac{\overline{v'\theta'}}{\overline{\theta}_z} - \overline{w'u'} \right\} \tag{2}$$

$$\nabla \cdot F \equiv (a\cos\varphi)^{-1} \frac{\partial}{\partial\varphi} (F_\varphi \cos\varphi) + \frac{\partial F_z}{\partial z} \tag{3}$$

Where $\rho_0$ is the density; $z$ is the altitude; $a$ is the radius of the Earth; $\varphi$ is the latitude; $f$ is the Coriolis parameter; $\theta$ is the potential temperature; $u$ and $v$ are the zonal and meridional winds, respectively; $w$ is the vertical velocity. The overbars represent the zonal average, and primes represent deviation from the zonal average.

Mass stream function $\overline{\chi}*$ associated with the residual mean circulation is defined as:

$$\frac{\partial \overline{\chi}*}{\partial\varphi} = \rho_0 a \cos\varphi \overline{w}* \tag{4}$$

Where $\overline{w}*$ is represented as:

$$\overline{w}* = \overline{w} + (a\cos\varphi)^{-1} \left( \frac{\cos\varphi \overline{v'\theta'}}{\overline{\theta}_z} \right)_\varphi \tag{5}$$

In the SH, the climatological mean mass stream function is negative.

### 2.2.3 The refraction index

We use the method developed by Harnik and Lindzen (2001) to divide the traditional refraction index ( $n_{ref}^2$ ) into vertical ($m^2$) and meridional ($l^2$) components by solving the quasi-geostrophic equation in the spherical coordinate:

$$\frac{\partial^2 \psi}{\partial z^2} + \frac{N^2}{f^2}\frac{\partial^2 \psi}{\partial y^2} + n_{ref}^2 \psi = 0 \tag{6}$$

Where $\psi$ is the weighted wave geopotential height, and $N^2$ is the Brunt-Väisälä frequency. By solving Eq. (6), the vertical and meridional components of the refraction index can be expressed as:

$$\mathrm{Re}\left(\frac{\psi_{zz}}{\psi}\right) = -m^2 \text{ and } \mathrm{Re}\left(\frac{\psi_{yy}}{\psi}\right) = -l^2 \tag{7}$$

And regions with a large refraction index facilitate wave-propagation.

### 2.2.4 The Lorenz energy cycle

The Lorenz energy cycle is used here to measure the atmospheric circulation changes. As described by Lorenz (1967) and Holton (1968), the diabatic heating generates mean available potential energy (PM), which is described as (Hu et al., 2004):

$$PM = \frac{c_p}{2}\int \gamma [<T>]''^2 \, dm \tag{8}$$

$$\gamma = -\left\{\frac{\theta}{T}\left[\left(\frac{R}{c_p}\right)p\right]\right\} \tag{9}$$

And then the baroclinic eddies transport warm air poleward, cold air equatorward, and transform the PM to eddy available potential energy (PE). At the same time, the PE is transformed into eddy kinetic energy (KE) by the vertical motions of the eddies, and the KE can be described as:

$$KE = \frac{1}{2}\int [<u^2>+<v^2>]dm + \frac{1}{2}\int [<u>^{*2}+<v>^{*2}]dm \tag{10}$$

The mean kinetic energy (KM), defined as the integration of $\overline{u}^2$, can be maintained primarily by the conversion from KE due to the correlation $\overline{u'v'}$ (Eq. 11). Additionally, it can also be converted into PM (Eq. 12).

$$C(KE, KM) = \int [<u'v'> + <u>^* <v>^*] \cos \varphi \left\{ \frac{\partial ([<u>]/\cos \varphi)}{a \partial \varphi} \right\} dm$$

$$+ \int [<v'^2> + <v>^{*2}] \frac{\partial [<v>]}{a \partial \varphi} dm$$

$$+ \int [<u'w'> + <u>^* <w>^*] \frac{\partial [<u>]}{\partial p} dm \qquad (11)$$

$$+ \int [<v'w'> + <v>^* <w>^*] \frac{\partial [<v>]}{\partial p} dm$$

$$- \int [<v>][<u'^2> + <u>^{*2}] \frac{[\tan \varphi]}{a} dm$$

$$C(PM, KM) = -\int [<v>] g \frac{\partial [<z>]}{a \partial \varphi} dm \qquad (12)$$

Where $T$ is the temperature, $c_p$ is the specific heat of air at constant pressure, $p$ is the pressure, $R$ is the gas constant for dry air, and $g$ is the gravitational acceleration. The $< >$ and $[ ]$ represent the time and zonal average, respectively. The $*$ and $'$ denote the departure from zonal mean and time average, respectively. The $''$ represents departure from meridional average. $C(A, B)$ represents the conversion from A to B. The $\int [s] dm$ is defined as $\int_z \int_y \int_x s \rho_0 \, dx \, dy \, dz$.

## 2.3 Model simulation

In this study, we use the Whole Atmosphere Community Climate Model (WACCM) in the Community Earth System Model version 2 (CESM2) to investigate the connection between the QBO signal and the Antarctic stratospheric polar vortex. The model has a horizontal resolution of 1.9° × 2.5° and a hybrid vertical coordinate with 70 levels from the surface to approximately 140 km (Gettelman et al., 2019). To avoid the impacts of initial fields and stochastic low-frequency climate fluctuations on the experimental results (Zhang et al., 2024), 20 ensemble experiments are conducted. These experiments use the same annual cycle of external forcings in 2000 (e.g., sea ice concentration, sea surface temperature, stratospheric ozone and other chemistry species, greenhouse gas, aerosols, and solar radiation) but with different initial fields. The nudging technology uses a dynamical-core-independent scheme (Davis et al., 2022).

$$F = -W \frac{(X - X_{ref})}{\tau} \qquad (13)$$

Where $X_{ref}$ is the meteorological value from the reanalysis datasets at the next update step, $\tau$ is the relaxation timescale (6 hours), and W is the nudging coefficients from 0 to 1. Stratospheric zonal winds, meridional winds, and temperatures from 1.2 to 103.3 hPa are nudged with a nudging coefficient of 1.0. This nudging coefficient decreases linearly to 0 from 103.3 hPa to 143.0 hPa. Tropical stratosphere (22°S to 22°N) is nudged to a coefficient of 1.0, with a transitional latitude of 22°N/S to 35°N/S. The stratospheric conditions mentioned above are nudged to the JRA-55 reanalysis, and each nudging experiment

runs from 1980 to 2022. Note that different sets of reanalysis data are used to force the CESM and perform other analysis. The consistency of results across both datasets indicates the robustness of the findings.

**3 Results**

**3.1 A better predictor of the Antarctic stratospheric polar vortex and ozone**

Here, we use the zonal-mean zonal wind at 60ºS and 70 hPa as an indicator of the strength of the Antarctic stratospheric polar vortex. Figure 1a indicates its time-lag correlations with the QBO index. Anstey and Shepherd (2014) reviewed the impact of QBO on high latitudes. There are stronger westerlies during the WQBO, although with weak statistical significance. Lecouffe
et al. (2022) suggested that the QBO can modulate the strength of the stratospheric polar vortex, which tends to be stronger and more persistent during the WQBO. Therefore, our analysis focuses on the positive correlations in Fig. 1a, which represent a stronger stratospheric polar vortex during the westerly QBO phase. The QBO-SH stratospheric polar vortex connection emerges in July and persists until austral late spring. The maximum correlations appear in November (i.e., the correlation between the July QBO and the stratospheric polar vortex in November), which is consistent with the previous studies (Baldwin
and Dunkerton, 1998; Yamashita et al., 2018). In this study, we focus on the earliest QBO signal in the Antarctic stratospheric polar vortex to determine the longest possible prediction lead time. Note that there is a maximum positive correlation between the July QBO index and the Antarctic stratospheric polar vortex strength 4 months later, suggesting that the QBO signal in austral winter influences the Antarctic stratospheric polar vortex in austral spring. During late austral spring, the duration of the Antarctic ozone hole is highly correlated with the strength of the polar vortex. Consequently, the winter QBO potentially
serves as a predictor of the Antarctic stratospheric polar vortex and the ozone hole in austral spring. However, this QBO-vortex signal is not statistically significant. We examine the response of the Antarctic stratospheric polar vortex in austral spring to the winter QBO (Fig. 1b). The WQBO in winter (QBO index in July greater than 1) does not consistently lead to larger zonal-mean zonal winds in the polar regions during spring, and the correction between them is only 0.23. These results suggest that the direct impact of the QBO on the Antarctic stratospheric polar vortex is weak.

To further explore the robust relationship between the winter QBO and the Antarctic stratospheric polar vortex in spring, we first regress the zonal-mean zonal winds in winter against the strength of the polar vortex in November during the WQBO. Interestingly, the winter zonal-mean zonal winds around 30ºS in the upper stratosphere highly correlate with the Antarctic stratospheric polar vortex in November (Figures 2a−c). Additionally, the winter mid-latitude winds also show high correlations with the Antarctic ozone in November (Figs. 2d−f). These results suggest that the extratropical zonal-mean zonal winds in the
upper stratosphere (hereafter, extratropical mode), along with the WQBO signal in winter, might serve as a joint predictor of the Antarctic stratospheric polar vortex and ozone in austral spring.

Here, the SVD analysis is further used to depict the relationship between the winter extratropical mode (Fig. 2a) and the Antarctic stratospheric polar vortex in spring. Figures 3a and b show the first paired mode of zonal mean zonal winds in the extratropical and polar regions, respectively. The extratropical mode is characterized by positive zonal wind anomalies

centered around 30°S, which closely resemble the regression patterns in Fig. 2. In the tropical regions, it displays a weak WQBO signal at 20 hPa (right panel of Fig. 3a). In austral spring, polar regions feature positive zonal wind anomalies south of 50°S, indicating a strong polar vortex (Fig. 3b). The first paired mode explains 98.2% of the total variance and is significant at the 95% confidence level according to the Monte Carlo test. Since the first paired mode represents the dominant coupled variance, our analysis mainly focuses on the first SVD paired mode. The correlation coefficient between the time series of the

two modes is 0.75, which is also significant at the 95% confidence level (Fig. 3c). Note that almost all years represented by the circles fall into the first and third quadrants in Fig. 3c, suggesting that when the extratropical mode is in its positive (negative) phase, a stronger (weaker) polar vortex will occur four months later. It is worth noting that the blue and yellow circles, representing the phase and strength of ENSO, show no uniform pattern with the strength of the polar vortex in November. This suggests that the strong correlation between the two modes appears to have little connection with ENSO.

Moreover, the Antarctic stratospheric ozone in austral spring, which is closely related to the stratospheric polar vortex, also exhibits a high correlation of 0.6 with the extratropical mode in July (Fig. 3c). Thus, we can conclude that the winter extratropical mode, in conjunction with the WQBO, is closely linked to the spring Antarctic stratospheric polar vortex and ozone.

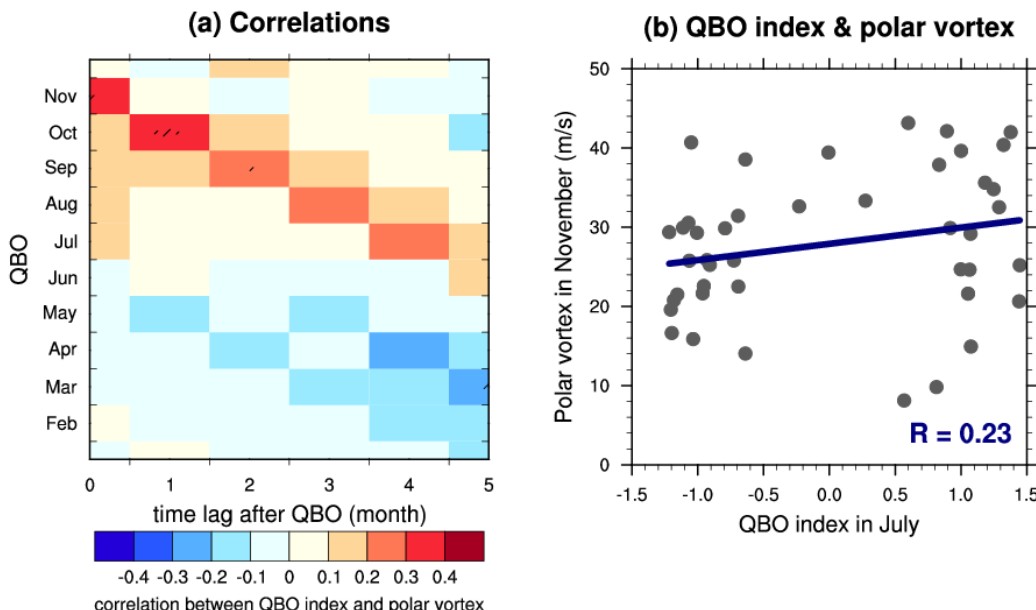


**Figure 1: (a) Correlations between the zonal-mean zonal wind at 60ºS and 70 hPa and the QBO index at different time lags derived from the MERRA-2 reanalysis dataset. The Y-axis represents the month of the QBO index used in the correlation analysis. The X-axis indicates the time lags. The shadings indicate that the correlations are statistically significant at the 90% confidence level. (b) Monthly mean zonal-mean zonal wind at 60ºS and 70 hPa in November (with a 5-month lag after the QBO in July) plotted against**
**the July QBO index according to the MERRA-2 reanalysis dataset from 1980 to 2022. The solid blue line represents the linear regression of the QBO index and the strength of the Antarctic stratospheric polar vortex, with their correlation coefficient shown in the bottom right-hand corner.**

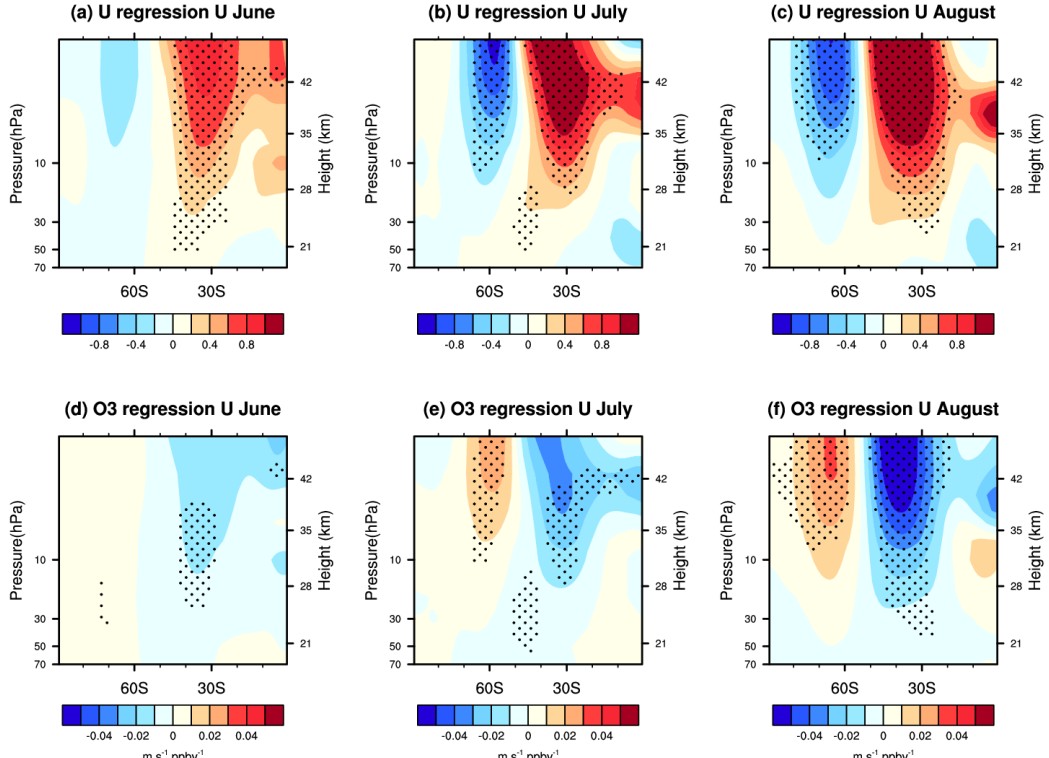

**Figure 2: (a)−(c)** Regression patterns of the zonal-mean zonal winds in (a) June, (b) July, and (c) August against the zonal-mean zonal wind at 60ºS and 70 hPa in November during the WQBO, derived from the MERRA-2 reanalysis dataset. **(d)−(f)** Regression patterns of the zonal-mean zonal winds in (d) June, (e) July, and (f) August against the ozone concentration averaged over the south of 60ºS at 70 hPa in November during the WQBO, derived from the MERRA-2 reanalysis dataset. The shadings indicate that the regression coefficients are statistically significant at the 95% confidence level.

### 3.2 The underlying mechanisms responsible for the relationship between the extratropical mode and the Antarctic stratospheric polar vortex during the WQBO

Note that the winter extratropical mode appears early in June and gradually develops (Fig. 2). In July, the extratropical mode is fully formed, and the correlation between the winter extratropical mode and the polar vortex in spring reaches its peak (0.74 in June, 0.75 in July, and 0.73 in August). Therefore, we primarily focus on the relationship between the extratropical mode in July and the Antarctic stratospheric polar vortex in November. As mentioned above, when the extratropical mode in July is in its positive phase (i.e., when the extratropical mode is greater than 0), a stronger polar vortex is observed four months later, and vice versa (Fig. 3c). Thus, first we categorize the WQBO into the Positive Extratropical mode (Pos-Exmode; 1980, 1990, 2006, 2008, 2015, 2022) and the Negative Extratropical mode (Neg-Exmode; 1985, 1997, 2004, 2013, 2016). The Pos-Exmode corresponds to a positive phase of the extratropical mode in July, while the Neg-Exmode corresponds to a negative phase (Fig. 3c). A composite analysis is then conducted to compare these two categories.

Figures 4a−e show the differences in composite monthly mean zonal-mean zonal winds between the Pos-Exmode and Neg-Exmode from July to November. In the upper stratosphere, a positive center of zonal wind anomalies is located at 20°S−40°S, while a negative center is located at higher latitudes in July. From July to November, the positive zonal-mean zonal wind anomalies exhibit a downward and poleward movement, resulting in a stronger polar vortex in austral spring. Meanwhile, the negative anomalies also shift towards higher latitudes and gradually weaken, eventually being replaced by the

positive anomalies in November. The evolution of these zonal-mean zonal wind anomalies confirms the connection between the extratropical mode in winter and the polar vortex in November during the WQBO. We further reveal that the positive zonal-mean zonal wind anomalies can be reasonably explained by anomalous E-P flux divergences and their poleward shift from July to November. The composite differences in E-P flux and its divergence between the Pos-Exmode and Neg-Exmode are shown in Figs. 4f−j. In July and August, anomalous E-P flux divergences (Figs. 4f−g) are established between 30°S and

60°S in the upper stratosphere, corresponding to weaker equatorward and upward propagation of planetary waves in these regions than normal. These E-P flux divergences lead to the poleward shift of the positive zonal-mean zonal wind anomalies from July to August (Figs. 4a−b). Note that the E-P flux divergence anomalies shift to 60°S in the mid and lower stratosphere in September, with these positive anomalies persisting until November.

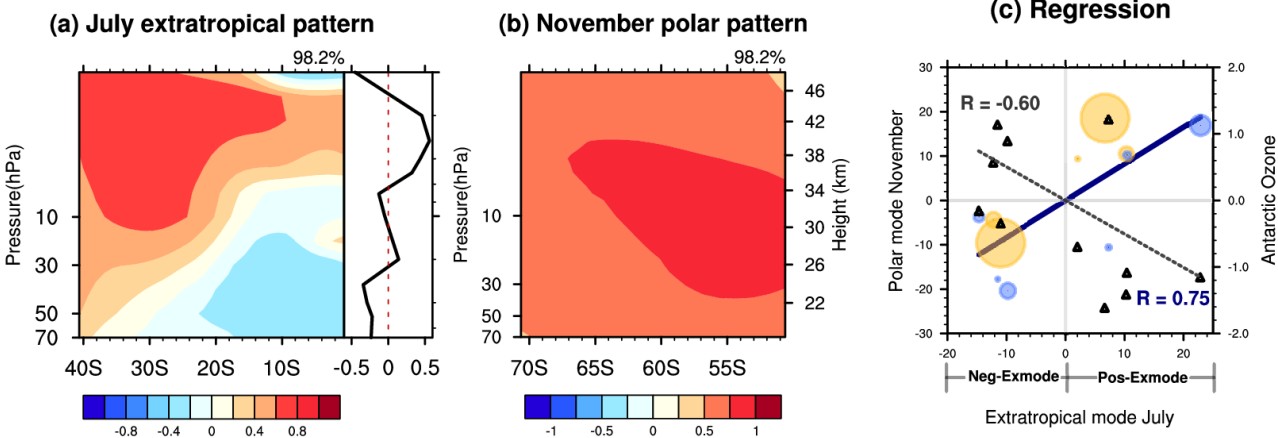

**Figure 3: Spatial patterns for the first paired mode of the (a) monthly mean zonal-mean zonal wind over 0−40°S and 1−70 hPa in July, i.e., extratropical mode (b) zonal-mean zonal wind over 50°S−70°S and 1−70 hPa in November by the singular value decomposition (SVD) analysis during the WQBO, based on the MERRA-2 reanalysis dataset from 1980 to 2022. The right panel of (a) is the profile of the extratropical mode averaged over 0−5°S. The variance explained by the first mode is shown at the top right-hand corner. (c) The corresponding time series for the paired mode, with their correlation coefficient shown at the bottom right-**
**hand corner (text in blue). The solid blue line represents the linear regression of the extratropical mode and polar mode. The size and color of the circle markers in panel (c) are proportional to the Niño 3.4 index, with yellow dots indicating positive Niño 3.4 indices and blue dots indicating negative Niño 3.4 indices. The standardized ozone volume mixing ratios, averaged over 60°S−90°S at 70 hPa in November, against the extratropical mode time series are shown in triangular markers (right Y-axis), with their correlation coefficient displayed at the top left-hand corner (text in black). The dashed black line represents the linear regression of**
**the extratropical mode and ozone in November.**

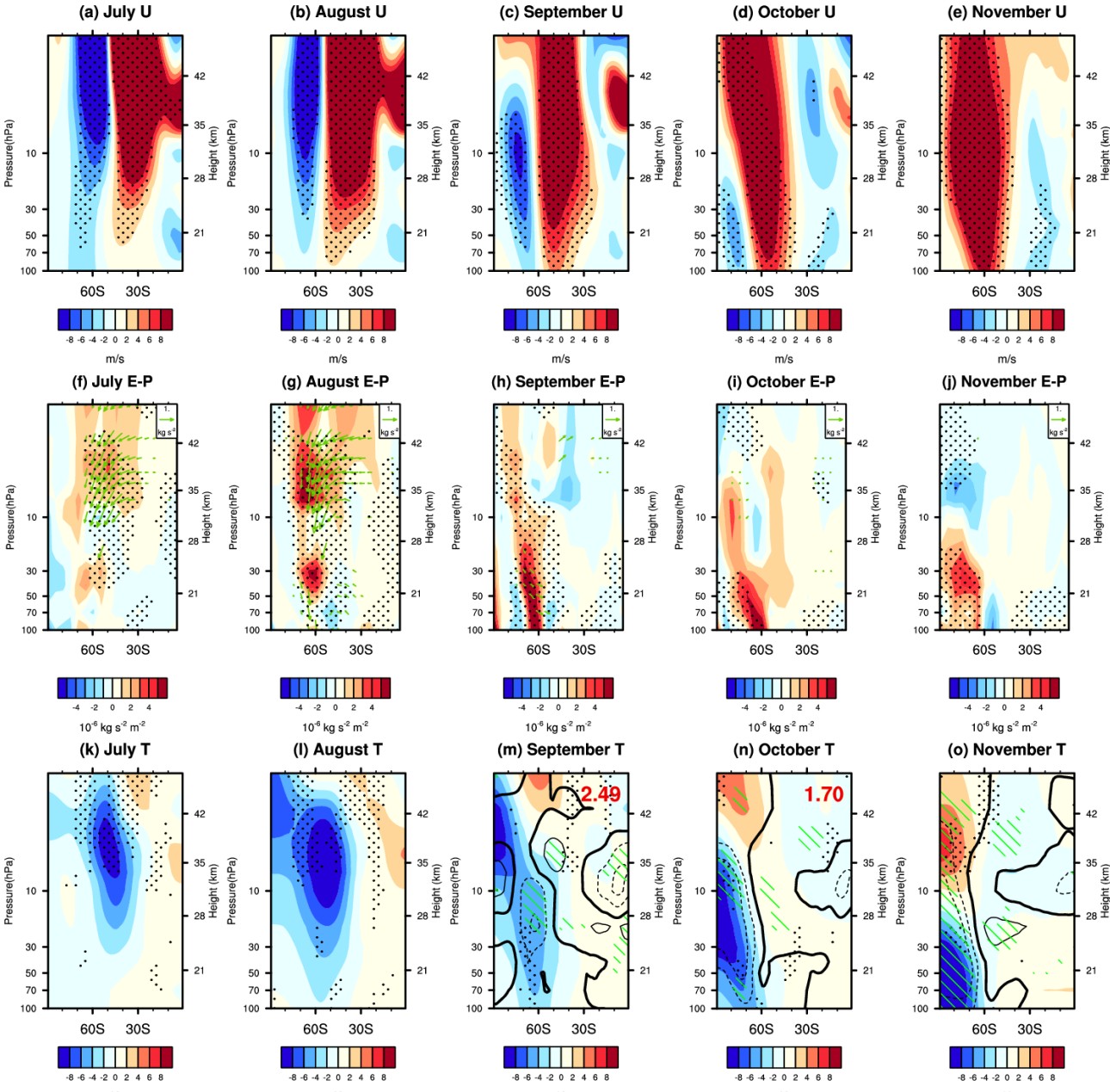

**Figure 4: Composite differences in (a)−(e) the zonal-mean zonal wind, (f)−(j) the scaled E-P flux (green vectors; horizontal component unit: $10^7$ kg s$^{-2}$; vertical component unit: $10^5$ kg s$^{-2}$) and the E-P flux divergence (shadings), (k)−(o) the zonal-mean temperature (shadings) and the zonal-mean ozone volume mixing ratio (contours; dashed lines are negative, and thick lines are zero contours. The contour intervals are 200 ppbv) from July to November between the Pos-Exmode and Neg-Exmode according to the MERRA-2 reanalysis dataset. Composite differences in PSC areas between the Pos-Exmode and Neg-Exmode, based on the NASA ozone watch dataset from September to October, are shown at the top of panels (m) and (n). The E-P flux and its divergence are calculated from wave 1 to 3. E-P flux vectors are scaled by the factor $\cos\varphi$ and multiplied by the square root of 1000.0/$p$ in both the**

vertical and horizontal directions, where *p* is the pressure in hPa. The dotted regions indicate that the differences in (a)−(e) zonal-mean zonal wind, (f)−(j) E-P flux divergence, and (k)−(o) zonal-mean temperature are statistically significant at the 90% confidence level. Green shadings indicate that the differences in ozone volume mixing ratio between the Pos-Exmode and Neg-Exmode are statistically significant at the 90% confidence level. Only the significant (at the 90% confidence level) E-P flux vectors have been plotted in panels (f)−(j).

Additionally, negative temperature anomalies are located south of the center of positive zonal wind anomalies according to the thermal wind balance. There is also a poleward and downward shift in temperature anomalies along with the anomalous zonal wind from July to November (Figs. 4k−o). Finally, the lower Antarctic stratosphere is much colder in the Pos-Exmode than Neg-Exmode. In austral spring (September to November), temperature plays a crucial role in influencing the Antarctic stratospheric ozone. Lower temperature in the lower stratosphere favours increased PSC area and chemical ozone depletion in

the polar regions (Figs. 4m−n). Consequently, the temperature decrease induced by the positive extratropical mode leads to negative ozone anomalies in the Antarctic lower stratosphere from September to November (Figs. 4m−o).

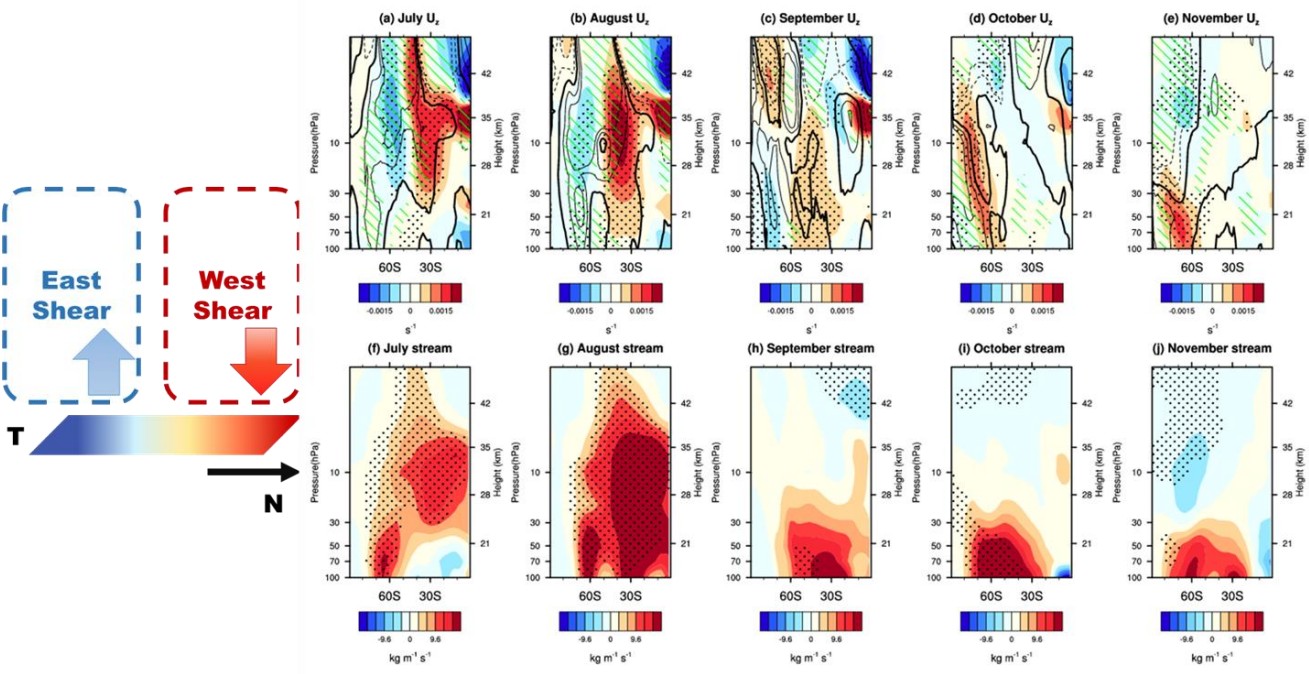

**Figure 5: Composite differences in the (a)−(e) vertical gradient of zonal-mean zonal wind (shadings) and vertical component of residual mean circulation ( $\overline{w}*$ ; contours; dashed lines are negative, and thick lines are zero contours. The contour intervals are 0.1**
**mm/s), and (f)−(j) stream function from July to November between the Pos-Exmode and Neg-Exmode according to the MERRA-2 reanalysis dataset. The dotted regions indicate that the differences in (a)−(e) vertical gradient of zonal-mean zonal wind and (f)−(j) stream function between the Pos-Exmode and Neg-Exmode are statistically significant at the 90% confidence level. Green shadings indicate that the differences in $\overline{w}*$ between the Pos-Exmode and Neg-Exmode are statistically significant at the 90% confidence level. The schematic diagram on the left illustrates how the secondary circulation is triggered and sustained.**

There are three main processes responsible for the extratropical-polar connection during the WQBO. The first process, named as the thermal wind balance, is described as follows. The positive extratropical mode defined in Fig. 3a exhibits positive zonal-mean zonal wind anomalies, as well as positive vertical gradient anomalies around 30ºS (Figure 5a). Meridional

temperature anomalies must exist to maintain the thermal wind balance for the vertical shear of the zonal-mean zonal winds in the basic state flow. These meridional temperature anomalies are sustained by secondary vertical motion. Specifically, a positive zonal-mean zonal wind shear anomaly (shadings in Fig. 5a), accompanied by a warm center to north of 30ºS (Fig. 4k), is maintained by downward motion (contours in Fig. 5a). While a negative zonal-mean zonal wind shear anomaly at higher latitudes is associated with a cold center and upward motion (Figs. 4k and 5a). Note that the climatological basic state meridional flow in austral spring is characterized by a negative stream function, with upward motion in the tropics and downward motion at higher latitudes (i.e., Brewer-Dobson circulation). In July and August, positive stream function anomalies appear at lower latitudes in the upper stratosphere, indicating clockwise secondary circulations induced by the extratropical mode during the WQBO (Figs. 5f−g).

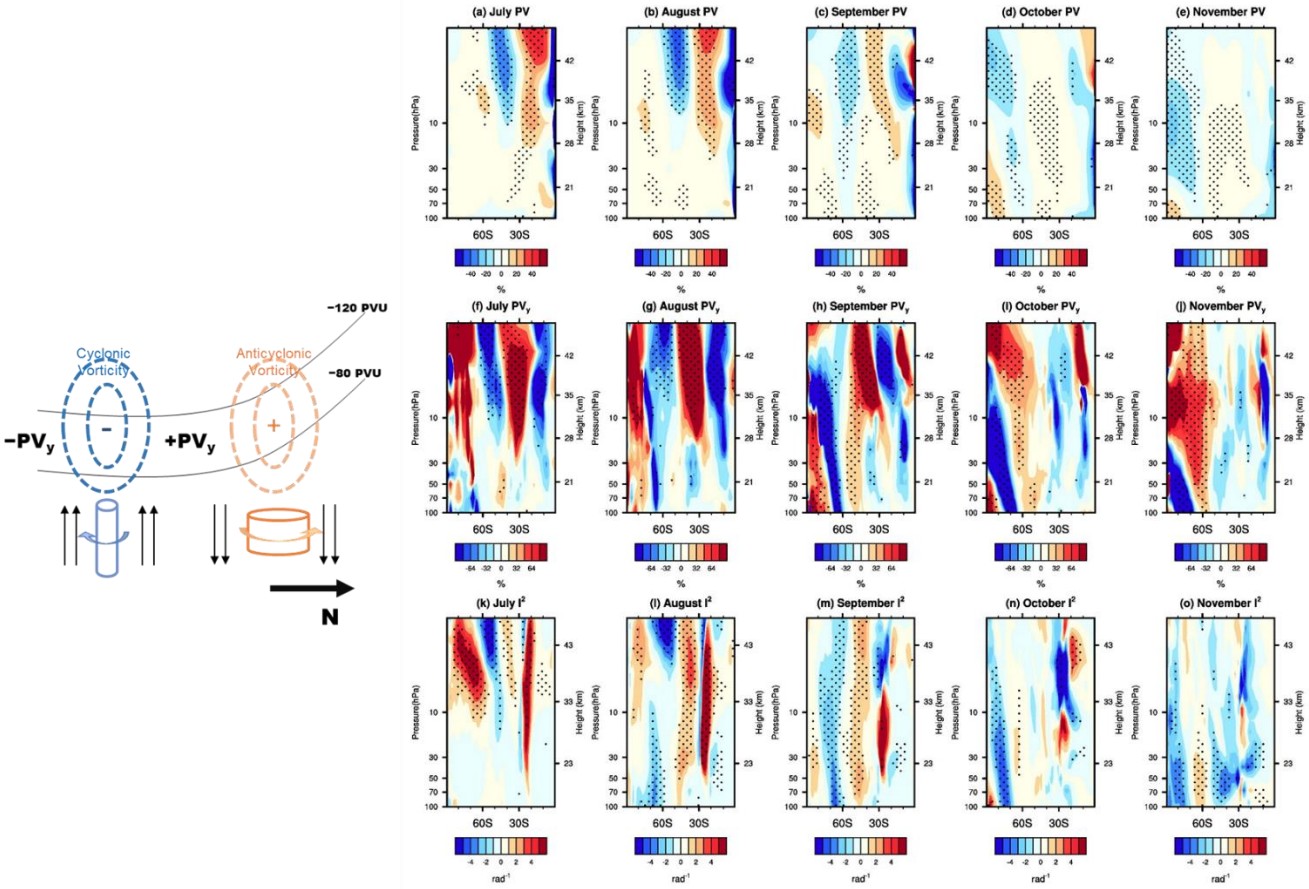

**Figure 6: Composite differences in the (a)−(e) potential vorticity (PV), (f)−(j) meridional gradient of PV, and (k)−(o) meridional component of refraction index from July to November between the Pos-Exmode and Neg-Exmode according to the MERRA-2 reanalysis dataset. The dotted regions indicate that the differences in (a)−(e) PV, (f)−(j) meridional gradient of PV, and (k)−(o) meridional component of refraction index between the Pos-Exmode and Neg-Exmode are statistically significant at the 90% confidence level. The schematic diagram on the left illustrates the mechanism of the PV adjustment. The black lines indicate the climatological PV contours in the SH.**

The second process is the potential vorticity (PV) adjustment. The secondary circulation mentioned above causes vertical compression and stretching of air columns in the lower and higher latitudes, respectively. As a result, the air columns will acquire anticyclone vorticity (positive PV anomalies) at lower latitudes and cyclone vorticity (negative PV anomalies) at higher latitudes (schematic diagram of Figure 6). In July, positive stream function anomalies are centered around 30ºS (Fig. 5f), resulting in positive PV anomalies at 20ºS and negative PV anomalies at 40ºS in the upper stratosphere (Fig. 6a). This
redistribution of PV leads to negative meridional gradient of PV ($PV_y$) anomalies around 60ºS (Fig. 6f), which dominate the wave refractive index and result in anomalous E-P flux divergence around 50ºS in the upper stratosphere in July (Fig. 4f). In the following months, the positive center of the stream function gradually shifts toward the lower stratosphere and polar regions (Figs. 5g−j). Consequently, the induced negative $PV_y$ anomalies and wave refractive index also exhibit a poleward and downward shift (Fig. 6). Especially from August to September, there is a notable transition in E-P flux divergence anomalies
from the upper stratosphere around 60ºS to the middle and lower stratosphere (30−100 hPa; Figs. 4g−h). At the same time, both the anomalous centers of the stream function and $PV_y$ also shift downward to the lower stratosphere (Figs. 5g−h, 6b−c, 6g−h, and 6l−m). This downward and poleward shift can be interpreted as wave-mean flow interaction, as described below.

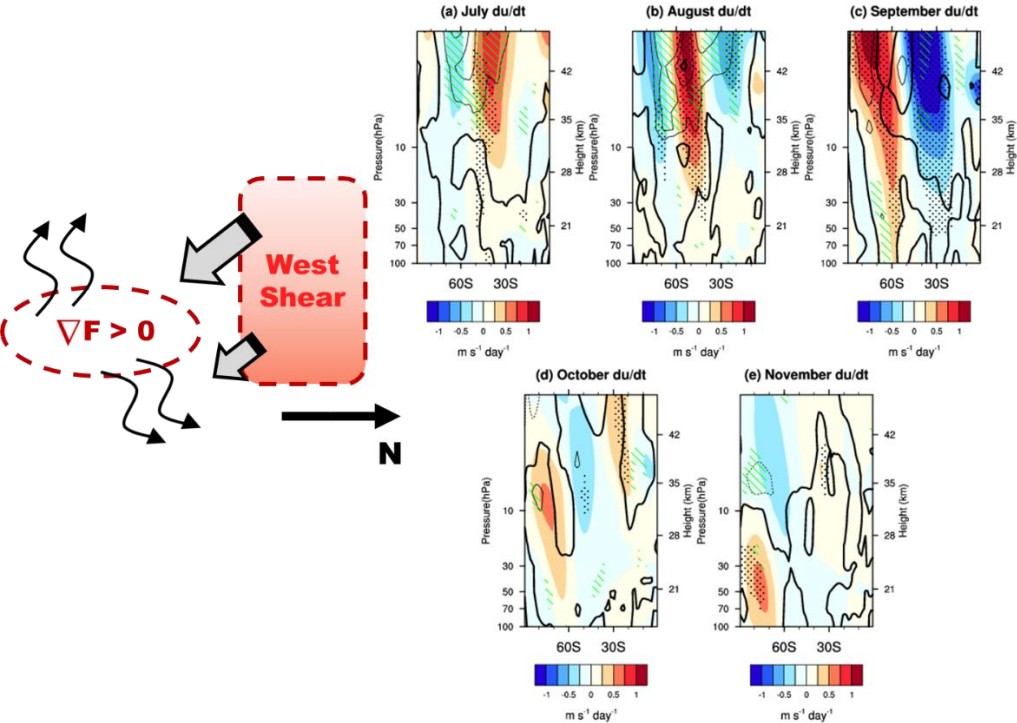

**Figure 7: Composite differences in the d$u$/d$t$ (color shadings) and E-P flux divergence (contours; dashed lines are negative, and thick**
**lines are zero contours. The contour intervals are 0.25 m s$^{-1}$ day$^{-1}$) from July to November between the Pos-Exmode and Neg-Exmode according to the MERRA-2 reanalysis dataset. The dotted regions and green shadings indicate that the differences in d$u$/d$t$ and the differences in E-P flux divergence between the Pos-Exmode and Neg-Exmode are statistically significant at the 90% confidence level. The schematic diagram illustrates how these anomalies propagate toward the Antarctic lower stratosphere from July to November.**

The wave-mean flow interaction sustains the anomalies mentioned above propagating toward the Antarctic lower
stratosphere from July to November (schematic diagram in Figure 7). Specifically, in July, the anomalous E-P flux divergence centered around 50ºS in the upper stratosphere (Fig. 7a) induces a poleward and downward shift of the positive anomalous zonal-mean zonal wind (positive anomalies of d$u$/d$t$), along with other anomalous centers. In the following months, the anomalous E-P flux divergence, as well as the positive d$u$/d$t$ anomalies, continues to move toward the lower stratosphere in the polar regions. By November, the center of the positive zonal-mean zonal wind anomaly is located in the polar regions of
the mid-stratosphere (Fig. 4e), indicating a stronger and colder polar vortex.

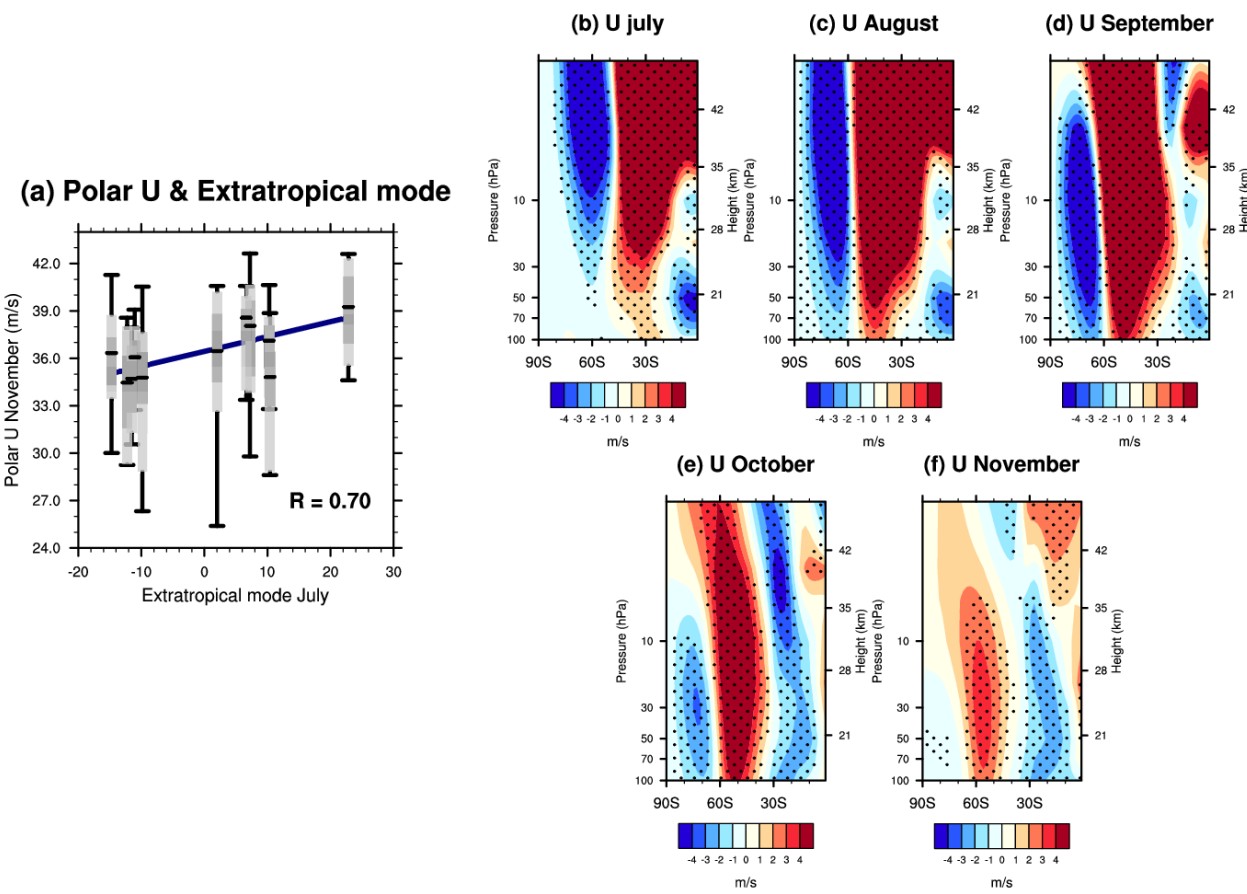

**Figure 8: (a) The zonal-mean zonal wind at 60ºS and 100 hPa in November is plotted against the extratropical mode's PC in July during the WQBO, derived from the CESM simulations. The boxplot describes a summary of the 20 ensembles. The light grey box spans from the lower decile to the upper decile, and the dark grey box spans the lower quartile to upper quartile. The lines inside**
**the dark grey box mark the median zonal wind. The lower and upper whiskers indicate the minimum and maximum zonal wind among the 20 ensembles. Additionally, the blue line represents the linear fit between the extratropical mode and the median zonal wind, with their correlation coefficient displayed at the bottom right-hand corner. (b)−(f) Composite differences in zonal-mean zonal wind from July to November between the Pos-Exmode and Neg-Exmode according to CESM simulations. The dotted regions indicate that the differences between the Pos-Exmode and Neg-Exmode are statistically significant at the 95% confidence level.**

340       Based on the abovementioned analyses, we propose a new predictor of the Antarctic stratospheric polar vortex in spring. During the WQBO, the correlation between the extratropical mode and the Antarctic stratospheric polar vortex can reach 0.75 with a five-month time lag. To further verify these results, additional model experiments are conducted by CESM2. First, the relationship between the extratropical mode in July and the Antarctic stratospheric polar vortex in November has been validated. Figure 8a shows the zonal-mean zonal wind at 60ºS in November as a function of the extratropical mode in July, derived from

the 20 ensembles during the WQBO. Similar to the reanalysis dataset (Fig. 3c), the strength of the Antarctic stratospheric polar vortex increases with a larger extratropical mode, with a correlation reaching 0.70. We further display the composite results of the zonal-mean zonal winds (Figs. 8b−f). For each ensemble, years in the Pos-Exmode and Neg-Exmode are selected, and the differences in zonal-mean zonal wind between the Pos-Exmode and Neg-Exmode are calculated across all ensembles. Initially, positive zonal-mean zonal wind anomalies appear around 30ºS in the upper stratosphere in July (Fig. 8b). These

anomalies gradually shift to the lower stratosphere and polar regions due to wave-mean flow interactions (Figs. 8c−e), ultimately resulting in a stronger polar vortex in the Antarctic lower stratosphere by November (Fig. 8f). The continuous evolution of these positive anomalous zonal-mean zonal winds from the extratropical upper stratosphere to the Antarctic lower stratosphere indicates a robust relationship between the extratropical mode and the Antarctic stratospheric polar vortex during the WQBO.

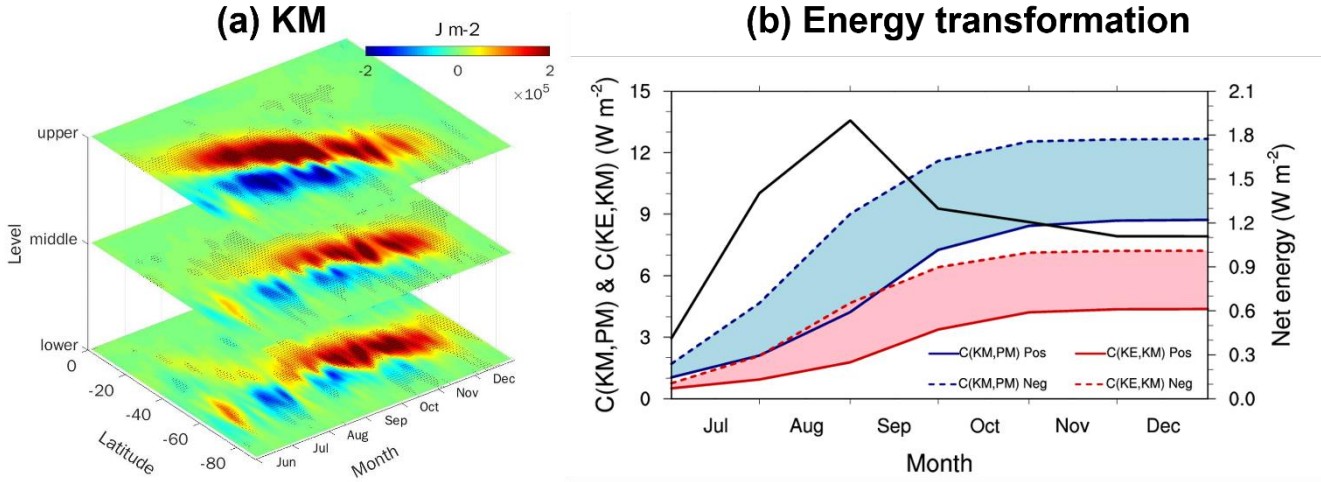


**Figure 9: (a) Composite differences in mean kinetic energy (KM) from July to November between the Pos-Exmode and Neg-Exmode according to the MERRA-2 reanalysis dataset. The KM at the "upper", "middle", and "lower" levels are integrated over 1−20 hPa, 30−50 hPa, and 70−100 hPa, respectively. Dotted regions indicate that the differences between the Pos-Exmode and Neg-Exmode are statistically significant at the 90% confidence level. (b) Composite time-integrated energy transformations averaged over 1−100**

**hPa and 40ºS−70ºS during the Pos-Exmode and Neg-Exmode. The blue and red shadings indicate the differences in C(KM, PM) and C(KE, KM) between the Pos-Exmode and Neg-Exmode, respectively. The black line indicates the net energy stored in KM. The blue and red lines correspond to the left Y-axis, while the black line corresponds to the right Y-axis.**

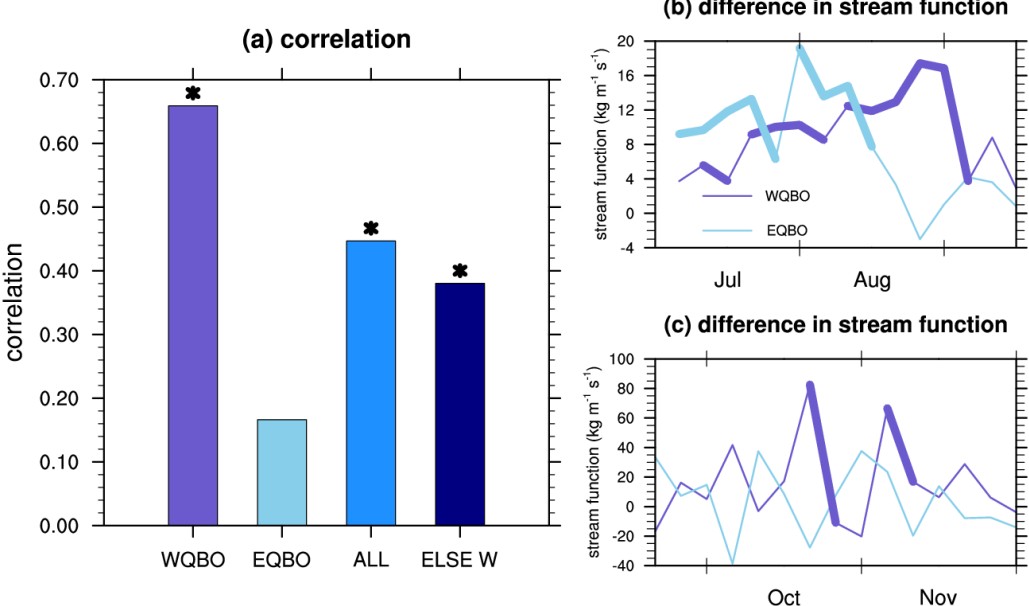

Figure 10: (a) Correlations between the zonal-mean zonal wind in extratropical and polar regions during the WQBO (WQBO), EQBO (EQBO), all years (ALL), and years excluding WQBO (ELSE W) derived from the MERRA-2 reanalysis dataset from 1980 to 2022. Asterisks indicate that the correlations are statistically significant at the 95% confidence level. (b) Composite differences in stream function averaged over 0−60ºS and 1−30 hPa between the positive and negative extratropical modes during the WQBO (purple lines) and EQBO (blue lines) from July to September. Thick lines indicate that the differences are statistically significant at the 90% confidence level. (c) Same as panel (b), but with the stream function averaged over 30ºS −60ºS and 50−100 hPa from September to November.

A question arises as to why the abovementioned wave-mean flow interactions could persist for almost five months. We attempt to explain it from the perspective of the Lorenz energy cycle. Lorenz (1967) proposed that solar energy is converted into available potential energy, which then drives kinetic energy to maintain the atmospheric circulation against both thermal and mechanical dissipations. As a result, the Lorenz energy cycle reflects changes in atmospheric circulation. Figure 9a displays the evolution of KM from June to December, which is proportional to the square of the zonal-mean zonal wind ($\overline{u}^{2}$). Therefore, KM can effectively capture the transition in atmospheric circulation depicted in Figs. 4a−e. Positive KM anomalies originate in the mid-latitudes of the upper stratosphere at the end of June. These anomalies propagate poleward and downward until November. This evolution of KM suggests a continuous anomalous energy transfer from the mid latitudes in the upper stratosphere to the polar regions, which sustains the positive zonal-mean zonal wind anomalies observed in the Pos-Exmode. Next, we explain why the positive anomalies in KM can persist for almost five months. Note that KM is primarily maintained by conversion from KE (Eq. 11) and PM (Eq. 12). In the SH, the climatological B-D circulation corresponds to the energy conversion from KM to PM (Li et al., 2007). The upward branch of the B-D circulation lifts relatively colder air at 20°S to higher altitudes, while relatively warmer air sinks at 60°S. This circulation raises the air mass center and results in energy conversion from KM to PM. Compared to the Neg-Exmode, tropical upward motion and mid-latitude downward motion are

weaker during the Pos-Exmode, as indicated by the anomalous upward motion at 60°S and downward motion in the tropical regions (Fig. 5a). This suggests that, during the Pos-Exmode, less KM is converted to PM, and as a result, more energy is stored in KM (Fig. 9b). In addition, KM is also maintained by conversion from KE. In the Pos-Exmode, weaker wave activities (Figs. 4f–j) result in less KE being converted into KM compared to the Neg-Exmode (red lines in Fig. 9b). Note that the blue shadings in Fig. 9b are wider than the red shadings, indicating that the anomalous conversion rate between PM and KM is greater than that between KE and KM. As a result, more energy is stored in KM during the Pos-Exmode than in the Neg-Exmode (black line in Fig. 9b).

Note that this relationship only exists during the WQBO. The correlation between the extratropical mode in July and the Antarctic stratospheric polar vortex in November can reach 0.65 during the WQBO, but is as low as 0.15 during the EQBO (Figure 10a). This is due to the intrinsic differences in the B-D circulation between the WQBO and EQBO. Compared to the EQBO, there are anomalous descending motions around 15°S at 10 hPa in the WQBO (Figure 11a), which facilitate the formation and maintenance of the secondary circulation (Figs. 5f–j). Therefore, during the WQBO, the secondary circulation, as well as other anomalous centers, propagates downward and poleward, especially from September to November. In contrast, during the EQBO, anomalous ascending motions in the tropics offset the secondary circulations induced by the extratropical mode. Thus, the secondary circulation weakens and gradually dissipates. Figs. 10b and c illustrate the composite differences in the spatial integral stream function. Note that at the beginning of July, the anomalous secondary circulation is stronger during the EQBO than during the WQBO. However, the extratropical-polar connection gradually disrupts from September in the EQBO.

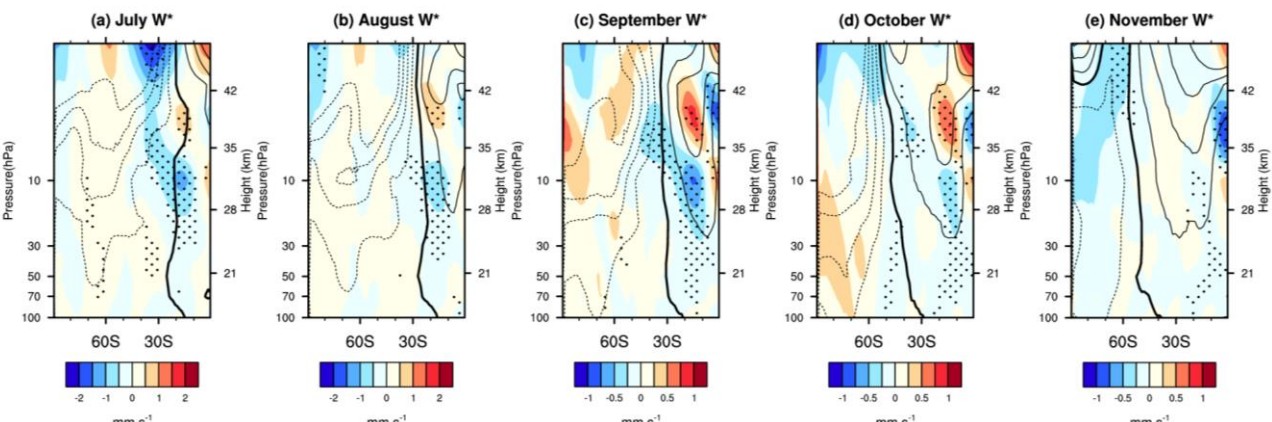

Figure 11: Composite differences in $\overline{w}*$ (color shadings) between the WQBO and EQBO from July to November according to the MERRA-2 reanalysis dataset. The dotted regions indicate that the differences in $\overline{w}*$ between the WQBO and EQBO are statistically significant at the 90% confidence level. The contours indicate the climatological $\overline{w}*$ from 1980 to 2022. The dashed lines are negative. Thick lines are zero contours, and the contour intervals are 0.5 mm/s.

## 4 Conclusion and discussion

The dynamical coupling between high and low latitudes has been widely discussed. However, no studies have pointed out a robust connection between the QBO and the Antarctic stratospheric polar vortex, as well as the Antarctic ozone. In this study, we use the MERRA-2 reanalysis data and CESM model simulations to investigate the relationship between the QBO and the Antarctic stratospheric polar vortex.

        During the WQBO, positive zonal-mean zonal wind anomalies at 20°S−40°S in the upper stratosphere in July, named as
the positive extratropical mode (Figs. 2 and 3a), lead to a stronger Antarctic stratospheric polar vortex in November, with a correlation reaching 0.75 (Fig. 3c). The positive extratropical mode can trigger a secondary circulation (Figs. 5f−j). The anomalous downward motion at lower latitudes and upward motion at higher latitudes cause an anomalous meridional gradient of PV (Figs. 6a−j), which further alters the environmental condition for wave propagation in the stratosphere (Figs. 6k−o). The resulting anomalous wave divergence leads to a stronger Antarctic stratospheric polar vortex in austral spring (Fig. 7). It takes
nearly five months for the positive zonal-mean zonal wind anomalies to propagate from 30°S to 60°S. Additionally, the anomalous Antarctic stratospheric polar vortex induced by the extratropical mode, along with the anomalous temperature in the Antarctic lower stratosphere, can influence the Antarctic ozone hole in austral spring (Figs. 4m−o). This results in a high correlation between the extratropical mode in July and the Antarctic ozone concentrations in November (Fig. 3c). Therefore, based on the relationship between the extratropical mode and the strength of the Antarctic stratospheric polar vortex, the
extratropical mode can be regarded as a predictor of the Antarctic polar vortex and ozone in austral spring during the WQBO. However, the reasons why the extratropical mode varies and its potential relationship with the QBO phase warrant further investigation.

        During the EQBO, the correlation between the extratropical mode and the strength of the polar vortex is only 0.15 (Fig. 10a). Due to stronger upward motion in the tropics, which opposes the secondary circulation caused by the extratropical mode,
the EQBO can only sustain the positive zonal-mean zonal wind anomalies until September. Further analysis is needed to find a realistic connection between the EQBO and the Antarctic stratospheric polar vortex.

        As mentioned in the literature review, the QBO impacts the stratospheric polar vortex through the middle and lower stratospheric pathway (Naoe and Shibata, 2010; Yamashita et al., 2011; 2018). In this study, we propose an upper stratospheric pathway for the QBO's impact on the Antarctic stratospheric polar vortex, which can persist for nearly five months. This
suggests that both the Antarctic stratospheric polar vortex and ozone concentrations in austral spring could be predicted up to five months in advance. Additionally, it is generally recognized that the EQBO has a greater influence on the stratospheric polar vortex than the WQBO. However, through the upper stratospheric pathway, the WQBO during austral winter, combined with the extratropical zonal-mean zonal winds in the upper stratosphere, is highly correlated with the Antarctic stratospheric polar vortex and ozone in austral spring.

**Data availability:** MERRA-2 data are available at https://disc.gsfc.nasa.gov/datasets?project=MERRA-2. The PSC area data can be obtained from https://ozonewatch.gsfc.nasa.gov/. The code used in this article is accessible from the corresponding author.

**Author contributions:** All authors designed the study. ZW analysed and prepared the data for the paper. JZ, ZW, DL, and SZ contributed to data interpretation and writing of the paper.

**Competing interests:** The authors declare that they have no conflict of interest.

**Acknowledgements:** We thank Dr. Yamashita, the editor, and the two anonymous reviewers for their helpful comments. This research is supported by the National Natural Science Foundation of China (U2442211, 42075062, 42130601). We thank the scientific teams at National Aeronautics and Space Administration (NASA) for providing the MERRA-2 reanalysis data. We thank the NCAR For the CESM2 model. We gratefully acknowledge Dr. Nili Harnik for providing the code of the quasigeostrophic model to calculate the index of refraction. We also appreciate the computing support provided by the Supercomputing Center of Lanzhou University.

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
