# Peer review of "The Joint Effect of Mid-latitude Winds and the Westerly Quasi-Biennial Oscillation Phase on the Antarctic Stratospheric Polar Vortex and Ozone"

_EGUsphere, 2024_

## Author Comment (AC1)

**Replies to Referee RC1**

**Manuscript number**: **EGUSPHERE-2024-2669**

**Title**: The Joint Effect of Mid-latitude Winds and the Westerly Quasi-Biennial Oscillation Phase on the Antarctic Stratospheric Polar Vortex and Ozone

**2024**

We thank the anonymous reviewer and the editor for your helpful comments which helped us greatly to improve our paper. We modified our paper according to the comments. Our replies are summarized as below:

In the original manuscript, we classified the WQBO into WQBO-Strong Polar Vortex (W-SPV) and WQBO-Weak Polar Vortex (W-WPV) according to the phase of the extratropical mode in July (Figure R1a). However, we realized that the two phases are misnamed, as a positive extratropical mode does not always lead to a stronger polar vortex, despite the strong correlation between them. In the revised manuscript, we renamed the positive and negative extratropic mode in July as Positive-Extratropic mode (Pos-Exmode) and Negative-Extratropic mode (Neg-Exmode) as in Figure R1b.

[Figure]

Figure R1. The corresponding time series for the extratropic mode and polar mode of Figure 3 in the manuscript. (a) The Figure 3c in the original manuscript. (b) The Figure 3c in the revised manuscript.

This paper establishes a robust connection between the QBO signal in winter and the stratospheric polar vortex in spring with a time-lag of five months. Their results indicated that zonal-mean zonal winds in the mid-latitude upper stratosphere play a crucial role in facilitating the tropic-polar connection in Southern Hemisphere.

Specifically, during WQBO, the positive zonal-mean zonal winds anomalies at 20°S−40°S in the upper stratosphere in July can lead to a stronger Antarctic stratospheric polar vortex and lower ozone concentrations in November. This finding on predicting the Antarctic polar vortex and ozone in spring could be of broad interest, and the authors have presented a comprehensive body of work on it. Overall, this paper presents an interesting and convincing and well-written analysis. I think this study would be of interest to the readership Atmospheric Chemistry and Physics and recommend its publication after addressing the comments listed below.

General comments:

In the introduction, the authors mentioned that most researches focus on the QBO-polar connection in the Northern Hemisphere (NH), where the upward-propagating planetary waves are strong. A more detailed explanation of the underlying mechanisms, along with a discussion of whether this connection in the NH is robust, would strengthen the introduction. I think this would offer readers more useful information about why there is less attention on the QBO-polar connection in the Southern Hemisphere.

**Response: Thank you for your comment. A detailed explanation of the Holton-Tan effect has been added in the revised manuscript.**

*"During the westerly QBO phase (WQBO), the zero-wind line of the zonal-mean zonal wind shifts equatorward, causing planetary wave to be reflected away from high latitudes. This dynamic is expected to strengthen the polar vortex by reducing wave-driven disturbances in the polar region. Consequently, the Arctic stratospheric polar vortex and the Brewer-Dobson circulation (hereafter referred to as B-D circulation) tend to be stronger on average during the WQBO compared to the easterly QBO phase (EQBO). Additionally, Garfinkel et al. (2012) proposed that the secondary meridional circulation induced by the EQBO also plays a crucial role in the Arctic stratosphere. This secondary circulation restricts the propagation of subpolar Rossby waves into the subtropics, resulting in more wave breaking closer*

*to the pole."*

The study defines the QBO phase using zonal-mean zonal wind at 20 hPa. However, I noticed that most researches define the QBO as being in its easterly (westerly) phase using the zonal mean zonal wind at 50 hPa. It would be very instructive to show why defining the QBO phase at 20 hPa is reasonable for establishing the QBO-polar connection in the SH.

**Response: Thank you for your comment. In the original manuscript, we discussed the reason why a single-level wind (20 hPa) has been used to select the WQBO and EQBO years. To clarify further, the empirical orthogonal function (EOF) analysis is applied to the tropical zonal-mean zonal wind, averaged over 10°S–10°N from 10 hPa to 70 hPa (following Randel et al., 1999; Anstey et al., 2010; Rao and Ren, 2018). The first EOF mode depicts the in-phase changes in equatorial stratospheric zonal wind from the lower to the upper stratosphere (black line in Figure R2a), accounting for 56.04% of the total variance and showing the maximum correlation with the single-level equatorial wind at 20–30 hPa. The second EOF mode represents the contrasting variations between the lower and upper stratosphere (black line in Figure R2b), and it agrees well with the equatorial wind at 50 hPa (blue line in Figure R2b) variation. However, the second EOF mode explains only about 30% of the total variance, suggesting that the first EOF mode-like equatorial wind at 20–30 hPa serves as a good indicator of the QBO phase.**

**Additionally, years in WQBO and EQBO are selected according to the equatorial wind at 20 and 30 hPa, as shown in Table R1. Note that the years identified by the 20 hPa zonal wind largely overlap with those at 30 hPa, except for 2022. Therefore, we use 20 hPa equatorial wind to define the QBO phase, consistent with previous studies (Baldwin et al., 2001; Naito, 2002; Rao et al., 2020).**

[Figure]

**Figure R2.** We first obtain the first two principal components (PCs) of the monthly mean zonal-mean zonal wind, averaged over 10°S–10°N from 10 hPa to 70 hPa in July, through empirical orthogonal function (EOF) analysis. (a) The vertical structure of the in-phase QBO mode (black line), and the correlation between this PC and the tropical zonal-mean zonal wind at different levels (blue line). (b) Same as panel (a), but for the out-of-phase mode.

**Table R1.** Years categorized as WQBO and EQBO are selected based on the tropical zonal wind at 20 hPa and 30 hPa.

| Level | WQBO |
|---|---|
| 20 hPa | 1980, 1985, 1990, 1997, 2004, 2006, 2008, 2013, 2015, 2016, **2022** |
| 30 hPa | 1980, 1985, 1990, 1995, 1997, 1999, 2002, 2004, 2006, 2008, 2013, 2015, 2016, 2019 |

Reference

Anstey, J. A., Shepherd, T. G. and Scinocca, J. F.: Influence of the quasi-biennial oscillation on the extratropical winter stratosphere in an atmospheric general circulation model and in reanalysis data, J. Atmos. Sci., 67, 1402–1419, doi: 10.1175/2009JAS3292.1, 2010.

Baldwin, M. P., Gray, L. J., Dunkerton, T. J., Hamilton, K, Haynes, P. H., Randel,

W. J., Holton, J. R., Alexander, M. J., Hirota, I., Horinouchi, T., Jones, D. B. A., Kinnersley, J. S., Marquardt, C., Sato, K. and Takahashi, M.: The quasi-biennial oscillation, Rev. Geophys., 39, 179–229, doi: 10.1029/1999RG000073, 2001.

Naito, Y.: Planetary wave diagnostics on the QBO effects on the deceleration of the polar-night jet in the southern hemisphere, J. Meteor. Soc. Japan, 80, 985–995, doi: 10.2151/JMSJ.80.985, 2002.

Rao, J., Garfinkel, C. I. and White, I. P.: Impact of the Quasi-Biennial Oscillation on the Northern Winter Stratospheric Polar Vortex in CMIP5/6 Models, J. Climate, 33, 4787–4813, doi: 10.1175/JCLI-D-19-0663.1, 2020.

Yamashita, Y., Naoe, H., Inoue, M. and Takahashi, M.: Response of the Southern Hemisphere Atmosphere to the Stratospheric Equatorial Quasi-Biennial Oscillation (QBO) from Winter to Early Summer, J. Meteorol. Soc. Jpn., 96, 6, 587−600, doi: 10.2151/jmsj.2018-057, 2018.

The correlation between the winter extratropical mode and the polar vortex reaches 0.75 during WQBO. It seems the winter extratropical mode could serve as a good predictor of the spring Antarctic stratospheric polar vortex. Additionally, in Figure 3c, a positive extratropical mode in July usually corresponds to a strong polar vortex. Would it be possible that the author uses these relationships to 'predict' the strength of the Antarctic polar vortex from 1950 to 1979 using the ERA5 reanalysis?

**Response: Thank you for your comment. Figure R3 shows the 'predicted' polar vortex during the WQBO phase from 1950 to 1979. Of the seven samples identified as WQBO, six are successfully classified to the strong or weak polar vortex by the extratropical mode, supporting the robustness of the mechanisms discussed in the original manuscript.**

[Figure]

**Figure R3. The standardized zonal-mean zonal wind at 60°S and 70 hPa plotted against the predicted polar vortex by the extratropical mode during the WQBO from 1950 to 1979 derived from ERA5 reanalysis dataset. The blue line represents the linear regression of the observation and predicted polar vortex.**

Specific comments:

Line 25 Please specify how QBO modify the upward-propagating planetary waves.
**Response: Thank you for your comment. The following sentence has been added to the revised manuscript.**
*"During the westerly QBO phase (WQBO), the zero-wind line of the zonal-mean zonal wind shifts equatorward, causing planetary wave to be reflected away from high latitudes. This dynamic is expected to strengthen the polar vortex by reducing wave-driven disturbances in the polar region. Consequently, the Arctic stratospheric polar vortex and the Brewer-Dobson circulation (hereafter referred to as B-D circulation) tend to be stronger on average during the WQBO compared to the easterly QBO phase (EQBO). Additionally, Garfinkel et al. (2012) proposed that the secondary meridional circulation induced by the EQBO also plays a crucial role in the Arctic stratosphere. This secondary circulation restricts the propagation of subpolar Rossby waves into the subtropics, resulting in more wave breaking closer*

*to the pole."*

Line 51 The QBO is only considered as a predictor of the Arctic stratospheric polar vortex and the near-surface climate in the NH. Please clarify this point.

**Response: Thank you for your comment. This sentence has been rewritten in the revised manuscript.**

*"The QBO period varies irregularly in the range from 17 to 38 months, which is considered as a reliable predictor of the stratospheric polar vortex, and further the near-surface climate and weather in the NH (Baldwin and Dunkerton, 2001; Zhang et al., 2020; Tian et al., 2023)."*

Line 74 Monthly to monthly

**Response: Thank you for your comment. Corrected.**

Line 77 what is the vertical range being considered?

**Response: Thank you for your comment. The vertical range of the SVD analysis has been clarified in the revised manuscript.**

*"The SVD analysis is performed between the zonal-mean zonal wind at latitudes ranging from 0° to 40°S and 1−70 hPa in July (extratropical mode) and the zonal-mean zonal wind at latitudes ranging from 50° to 70°S and 1−70 hPa in November (polar mode)."*

Line 96 Why not use the traditional refraction index to diagnose the wave-propagation in the stratosphere?

**Response: Thank you for your comment. First, note that from July to August, the difference in horizontal planetary wave between the Pos-Exmode and Neg-Exmode leads to anomalous E-P flux divergence (Figures 4f−g in the revised manuscript). Therefore, we display the meridional components of the refraction index (RI) in Figures 6k−l of the manuscript.**

Secondly, Figure R4 shows the difference in traditional RI between the Pos-Exmode and Neg-Exmode in July. Similar to the meridional components of the refraction index, there are negative RI anomalies between 55°S and 65°S in the upper stratosphere, suggesting that the refraction index used in the manuscript is appropriate for investigating the wave propagation environment.

[Figure]

**Figure R4. Composited difference in traditional RI between the Pos-Exmode and Neg-Exmode in July derived from MERRA-2 reanalysis dataset.**

Line 102 The word size of the equation (7) is too large. Please correct.

**Response: Thank you for your comment. Corrected.**

Line 136 I note that in the CESM, the stratospheric conditions are nudged to the JRA-55 reanalysis. Why is the model forced using different types of reanalysis data?

**Response: Thank you for your comment. We agree that discrepancies may exist between different reanalysis dataset, and validating the results across multiple datasets can enhance their robustness. First, we present the WQBO years selected based on the MERRA-2 and JRA55 reanalysis datasets in Table R2. The WQBO years show strong consistence between these two datasets.**

**Secondly, we reprinted Figure 3 of the manuscript using the JRA55 reanalysis dataset (Figure R5). This analysis similarly shows a strong correlation between**

the extratropic mode in July and the Antarctic polar vortex in late austral spring, suggesting that the conclusions in the original manuscript are not influence by the choice of reanalysis dataset. Additionally, in the original manuscript, we clarified the use of different reanalysis datasets, and the consistency of the results across both datasets further demonstrates the robustness of our findings.

[Figure]

**Figure R5. Same as Figure 3 of the manuscript, but it derived from JRA55 reanalysis datasets.**

**Table R2. Years categorized as WQBO based on the MERRA-2 and JRA55 reanalysis datasets.**

| Dataset | WQBO |
|---------|------|
| MERRA-2 | 1980, 1985, 1990, 1997, 2004, 2006, 2008, 2013, 2015, 2016, 2022 |
| JRA55 | 1980, 1985, 1990, 1997, 2004, 2006, 2008, 2013, 2015, 2016, 2022 |

Figure 1b It seems no apparent connection between the QBO in July and the Antarctic stratospheric polar vortex in austral spring. However, in the introduction, how previous studies have shown that the QBO can modulate the Antarctic polar vortex during austral spring?

**Response: Thank you for your comment. Some of the previous studies used the composite analysis to explore the connection between the QBO and the Antarctic**

stratospheric polar vortex (Baldwin and Dunkerton, 1998; Anstey and Shepherd, 2014). However, their composite analysis indicated only a slight response of the vortex to the QBO phase, and no statistical test to confirm these findings. Here, we also present the composited difference in zonal-mean zonal wind between the WQBO and EQBO. Even when using the November QBO index to estimate stratospheric Antarctic polar vortex for the same month, the significant difference in zonal-mean zonal wind between the WQBO and EQBO remains limited in extent, centered around 60°S at 100 hPa (Figure R6a). When the QBO signal leads to the stratospheric Antarctic polar vortex for almost five months, there is no significant difference in zonal-mean zonal wind between the WQBO and EQBO (Figure R6b).

However, Yamashita et al. (2018) examined the influence of the QBO on SH extratropical circulation from austral winter to early summer using a multiple linear regression approach. Their results are statistically significant. Thus, the mechanism in previous studies may not be adequate to explain the entirety of the extratropical response in the SH.

[Figure]

Figure R6. (a) Composite differences in zonal-mean zonal wind anomalies in November between the WQBO and EQBO according to MERRA-2 reanalysis

**dataset from 1980 to 2022. The phase of QBO is defined by using equatorial wind data in November. (b) Same as panel (a), but the phase of QBO is defined by using equatorial wind data in July. The dotted regions mark the differences in zonal-mean zonal wind are statistically significant at the 95% confidence level.**

Figures 3a and 3b The first paired mode explains 98.2% of the total variance. How is it possible for the first mode to account for nearly all of the total variance?

**Response: Thank you for your comment. Figure R7 presents the July extratropical pattern and the difference in zonal-mean zonal wind between the Pos-Exmode and Neg-Exmode. The two patterns are nearly identical, indicating that the extratropical pattern captures most of the information on the zonal-mean zonal wind distribution during the WQBO. Additionally, note that the November polar patter shown in Figure 3b of the manuscript represents the Antarctic polar vortex, the dominant feature of the polar region in austral spring. Consequently, the first paired mode accounts for nearly all of the total variance.**

[Figure]

**Figure R7. (a) Spatial patterns for the first paired mode of the (a) monthly mean zonal-mean zonal wind over 0−40°S and 1−70 hPa in July by the singular value decomposition (SVD) analysis during WQBO years, based on the MERRA-2 reanalysis dataset from 1980 to 2022. (b) Composite differences in the zonal-mean zonal wind anomalies in July between the Pos-Exmode and Neg-Exmode according to MERRA-2 reanalysis dataset.**

Figure 4 'horizontal component unit: 107 kg s−2; vertical component unit: 105 kg s−2'. Please correct the units.

**Response: Thank you for your comment. Corrected.**

Line 249 W* to w*

**Response: Thank you for your comment. Corrected.**

Line 306 '25 ensembles' to 20 ensembles

**Response: Thank you for your comment. Corrected.**

Line 381 positive anomalies in the zonal-mean zonal wind -> positive zonal-mean zonal wind anomalies

**Response: Thank you for your comment. Corrected.**

Line 400 'EQBO has a greater influence on the stratospheric polar vortex than the WQBO' Please explain.

**Response: Thank you for your comment. Figure R8 shows the probability distribution of the zonal-mean zonal wind at 60°S and 50 hPa in November. Compared to the black line, the green line shifts to the left, while the center position of the orange line aligns closely with the black line, suggesting that the EQBO has a greater influence on the Antarctic stratospheric polar vortex than the WQBO.**

[Figure]

**Figure R8. Probability distribution of 60°S zonal-mean zonal wind at 50 hPa in November during 1980−2022 (black line), WQBO (orange line), and EQBO (green line).**

---

## Author Comment (AC2)

**Replies to Referee RC2**

**Manuscript number**: **EGUSPHERE-2024-2669**

**Title**: The Joint Effect of Mid-latitude Winds and the Westerly Quasi-Biennial Oscillation Phase on the Antarctic Stratospheric Polar Vortex and Ozone

**2024**

We thank the anonymous reviewer and the editor for your helpful comments which helped us greatly to improve our paper. We modified our paper according to the comments. Our replies are summarized as below:

In the original manuscript, we classified the WQBO into WQBO-Strong Polar Vortex (W-SPV) and WQBO-Weak Polar Vortex (W-WPV) according to the phase of the extratropical mode in July (Figure R1a). However, we realized that the two phases are misnamed, as a positive extratropical mode does not always lead to a stronger polar vortex, despite the strong correlation between them. In the revised manuscript, we renamed the positive and negative extratropic mode in July as Positive-Extratropic mode (Pos-Exmode) and Negative-Extratropic mode (Neg-Exmode) as in Figure R1b.

[Figure]

**Figure R1. The corresponding time series for the extratropic mode and polar mode of Figure 3 in the manuscript. (a) The Figure 3c in the original manuscript. (b) The Figure 3c in the revised manuscript.**

**Summary**

Using the reanalysis and model simulations, this study analyzes the possible impact of the subtropical stratospheric wind mode on the QBO-Southern Hemisphere stratospheric polar vortex. The authors find that the westerly winds in the subtropics

can increase the correlation between the QBO and the polar vortex wind, while this relation is weak during the easterly winds in the subtropics. This finding is very interesting and important for seasonal forecast in the Arctic circulation and ozone. Therefore, I suggest to publish this paper with the following comments considered.

**Specific comments**

This study finds that the subtropical westerlies can increase the relationship between the WQBO and the strong polar vortex. However, this relationship is asymmetric. I meant that if the subtropical easterlies can increase the relationship between the EQBO and the weak polar vortex.

**Response: Thank you for your comment. First, the QBO-polar vortex coupling between the easterly quasi-biennial oscillation phase (EQBO) and westerly quasi-biennial oscillation phase (WQBO) is inherently asymmetric. Figure R2 shows the probability distribution of the zonal-mean zonal wind at 60°S and 50 hPa in November. Compared to the black line, the green line shifts to the left, while the center position of the orange line aligns closely with the black line, suggesting that the EQBO has a greater influence on the Antarctic stratospheric polar vortex than the WQBO.**

[Figure]

**Figure R2. Probability distribution of 60°S zonal-mean zonal wind at 50 hPa in November during 1980−2022 (black line), WQBO (orange line), and EQBO (green line).**

Furthermore, we agree that the relationship in our manuscript is also asymmetric between the WQBO and EQBO. As in the manuscript, we first show the regressed zonal-mean zonal wind in austral winter (from June to August) against the strength of the polar vortex in October and November during the EQBO. In EQBO, no uniform, statistically significant correlation pattern is observed throughout the winter in the upper stratospheric extratropical regions (Figures R3d−f), even with a shorter time lag (Figures R3a−c). Note that the most significant correlations appear in Figure R3c, where the polar vortex in October is related to the August zonal-mean zonal winds between 30°S−60°S in the upper stratosphere. This suggests that during the EQBO, the extratropic-polar connection to begin later, with the extratropical zonal wind positioned at higher latitudes. Given that, we replotted Figure 4 of the manuscript, but during the EQBO. Unlike in the WQBO, the extratropical mode in EQBO is calculated from the September zonal-mean zonal wind over 30°S−50°S and 1−70 hPa, with a correlation to the November Antarctic polar vortex reaching as high as 0.71. Thus, it's true that the relationship mentioned in our manuscript is asymmetric between the WQBO and EQBO.

[Figure]

**Figure R3. (a)−(c) Regression patterns of the zonal-mean zonal wind in (a) June, (b) July, and (c) August against the zonal-mean zonal wind at 60°S and 70 hPa in October derived from the MERRA-2 reanalysis dataset. (d)−(f) Regression patterns of the zonal-mean zonal wind in (d) June, (e) July, and (f) August against the zonal-mean zonal wind at 60°S and 70 hPa in November derived from the MERRA-2 reanalysis dataset. The shadings indicate that the regression coefficients are statistically significant at 95% confidence level.**

[Figure]

**Figure R4. Spatial patterns for the first paired mode of the (a) monthly mean zonal-mean zonal wind over 30−50°S and 1−70 hPa in September, (b) zonal-mean zonal wind over 50°S−70°S and 1−70 hPa in November by the singular value decomposition (SVD) analysis during EQBO years, based on the MERRA-2 reanalysis dataset from 1980 to 2022. (c) The corresponding time series for the paired mode, with their correlation coefficient shown in the bottom right-hand corner. The solid blue line represents the linear regression of the extratropical mode and polar mode. The size and color of the circle markers in panel (c) are proportional to the Niño 3.4 index, with yellow dots indicating a positive Niño 3.4 index and blue dots indicating a negative Niño 3.4 index.**

This study states that few studies focus on the relationship between QBO and the SH stratospheric polar vortex, and that the relation between them might be absent and non-existent. However, previous studies have reported the possible impact of QBO on the SH stratospheric polar vortex. The maximized response of the SH stratosphere to the QBO appears in boreal spring, not in winter (Rao et al. 2023a, b).

 **Response: Thank you for your comment. We agree the original phrasing was unclear, and we have revised the sentence as follows.**

*"In the Southern Hemisphere (SH), upward-propagating planetary waves are weak due to the weaker thermal contrast between land and sea. Consequently, the QBO-vortex coupling, which is closely related to planetary waves, has received less attention than those in the Northern Hemisphere (Garcia and Solomon, 1987; Lait et al., 1989; Baldwin and Dunkerton, 1998; Naito, 2002; Hitchman and Huesmann, 2009; Yamashita et al. 2018; Rao et al., 2023a, 2023b)."*

**Furthermore, in addition to the Naito et al. (2002) and Anstey et al. (2014), two other studies have been incorporated into the introduction.**

*"Yamashita et al. (2018) examined the influence of the QBO on SH extratropical circulation from austral winter to early summer using a multiple linear regression*

*approach. Their findings suggest that the QBO-SH polar vortex connection operates through two distinct pathways: the mid-stratospheric pathway, which tends to suppress the propagation of planetary waves into the stratosphere during WQBO, and the low-stratospheric pathway, which tends to enhance upward planetary waves in EQBO. The QBO-SH polar vortex connections established by Yamashita et al. (2018) are statistically significant. However, in QBO-resolving models from phases 5/6 of the Coupled Model Intercomparison Project (CMIP5/6), fewer than half of the General Circulation Models (GCM) successfully reproduce a weakened SH polar vortex during EQBO (Rao et al., 2023a). Furthermore, they also suggest that even the high-skill models capture only about 30% of the observed deceleration in westerlies during the EQBO. Although previous studies have revealed the potential relationship and mechanisms linking the QBO with the Antarctic stratospheric polar vortex, the weak statistical correlation between them and the limited performance of GCMs indicate that the QBO-vortex coupling in the SH is not jet fully understood."*

This study used model simulation by nudging methods. However, the method does not describe the necessity of the experiment. All figure captions should also mention if the results are based on ERA5 or ERA5. If the stratosphere is nudging, what can we learn from the experiments with t this study still focusing on the stratospheric variability?

**Response: Thank you for your comment. All figure captions have been verified, and the data source have been clearly specified.**

**The nudging simulation allows us to create additional WQBO ensemble members with varying initial conditions, reducing noise from the initial fields. In these model simulations, only tropical meteorological fields are nudged, while the other stratospheric regions evolve freely. This setup enables us to interpret the resulting extratropical patterns as responses to the WQBO signals in the tropical stratosphere (Figure R5).**

[Figure]

**Figure R5.** (a), (c), (e), (g), (i), and (k) Difference in temperature between the CESM model simulation and JRA55 reanalysis dataset at 50 hPa on 1 September 2021 from different ensembles. (b), (d), (f), (h), (j), and (l) Difference in zonal wind between the CESM model simulation and JRA55 reanalysis dataset on 1 September 2021.

Because the Whole Atmosphere Community Climate Model (WACCM) cannot reproduce a realistic QBO and instead produces weak easterlies in the equatorial lower stratosphere. Therefore, similar to most studies on the QBO, the equatorial zonal winds are relaxed toward idealized wind patterns (Matthes et al., 2010; Wang et al., 2022; Luo et al., 2024; Zhang et al., 2024). Zhang et al. (2024) used the QBO-nudging simulation to analyze precipitation in East Asia. Their findings further validate the effectiveness of nudging simulations for analyzing QBO influences on other processes. We used the same experiments derived from Zhang et al. (2024).

**Reference**

Luo, J. L. et al.: The impact of the QBO vertical structure on June extreme high temperatures in South Asia. npj Clim. Atmos. Sci., 7, 236, doi: 10.1038/s41612-024-00791-2, 2024

Matthes, K. et al.: Role of the QBO in modulating the influence of the 11 year solar cycle on the atmosphere using constant forcings. J. Geophys. Res., 115, D18110, doi: 10.1029/2009JD013020, 2010.

Wang, W. K., Hong, J., Shangguan, M., Wang, H. Y., Jiang W. and Zhao S. Y.: Zonally asymmetric influences of the quasi-biennial oscillation on stratospheric ozone, Atmos. Chem. Phys., 22, 13695–13711, doi: 10.5194/acp-22-13695-2022, 2022.

Zhang, R. H., Zhou W., Tian, W. S., Zhang, Y., Zhang, J. X. and Luo, J. L.: A stratospheric precursor of East Asian summer droughts and floods, Nat. Commun., 15, 247, doi: 10.1038/s41467-023-44445-y, 2024.

L15: anomalous zonal-mean zonal wind => zonal-mean zonal wind anomalies

**Response: Thank you for the careful check. It has been revised.**

L21: References are required for this sentence.

**Response: Thank you for your comment. We have revised this sentence as:**

*"The quasi-biennial oscillation (QBO) is a dominant mode of interannual variability in the tropical stratosphere (Lindzen and Holton 1968; Andrews and McIntyre 1976; Baldwin et al. 2001; Anstey and Shepherd, 2014; Rao et al., 2019; 2020a; 2023a)."*

L22: wind … descend => wind … descends

**Response: Thank you for the careful check. It has been revised.**

L31: Arctic ozone => Arctic ozone and water vapor

(https://doi.org/10.1016/j.wace.2023.100627)

**Response: Thank you for your comment. We have added 'water vapor'.**

*"This QBO-induced changes in the Arctic stratospheric polar vortex can further influence the distribution of Arctic ozone and water vapor (Wang et al., 2022; Lu et al., 2023)."*

L33: It depends on the timescale concerned. On the interannual timescale, the chemical processes are weaker than transport.

**Response: Thank you for your comment. This sentence has been written as:**

*"Zhang et al. (2021) revealed that dynamical processes contribute more to the Arctic ozone-QBO connection than chemical processes, although the impact of chemical processes on the Arctic ozone QBO signal is not negligible."*

L36: less thermal contrast => weaker thermal contrast

**Response: Thank you for the careful check. It has been revised.**

L38, L41-44: Two recent publications mentioned the impact of QBO on the Southern Hemisphere stratosphere.

**Response: Thank you for your comment. The two recent studies have been added in the revised manuscript.**

*"In the Southern Hemisphere (SH), upward-propagating planetary waves are weak due to the weaker thermal contrast between land and sea. Consequently, the QBO-vortex coupling, which is closely related to planetary waves, has received less attention than those in the Northern Hemisphere (Garcia and Solomon, 1987; Lait et al., 1989; Baldwin and Dunkerton, 1998; Naito, 2002; Hitchman and Huesmann, 2009; Yamashita et al. 2018; Rao et al., 2023a, 2023b)."*

*"Yamashita et al. (2018) examined the influence of the QBO on SH extratropical circulation from austral winter to early summer using a multiple linear regression approach. Their findings suggest that the QBO-SH polar vortex connection operates through two distinct pathways: the mid-stratospheric pathway, which tends to delay the downward evolution of the polar-night jet during WQBO, and the low-stratospheric pathway, which tends to enhance upward planetary waves in EQBO. The QBO-SH polar vortex connections established by Yamashita et al. (2018) are statistically significant. However, in QBO-resolving models from phases 5/6 of the Coupled Model Intercomparison Project (CMIP5/6), fewer than half of the General Circulation Models (GCM) successfully reproduce a weakened SH polar vortex during EQBO (Rao et al., 2023a). Furthermore, they also suggest that even the high-skill models capture only about 30% of the observed deceleration in westerlies during the EQBO. Although previous studies have revealed the potential relationship and mechanisms linking the QBO with the Antarctic stratospheric polar vortex, the weak statistical correlation between them and the limited performance of GCMs indicate that the QBO-vortex coupling in the SH is not jet fully understood."*

L48: It is too certain to say so.

**Response: Thank you for your comment. We agree that the phrase 'the only region' in the original manuscript seems too certain. We have revised the sentence as follows:**

*"Wang et al. (2022) suggested that the Antarctic total column ozone (TCO) shows no significantly respond to the QBO signal (Figure 2 in Wang et al., 2022)."*

L50: The QBO of which period varies irregularly in the range from 17 to 38 months is considered as a reliable predictor => The QBO period varies irregularly in the range from 17 to 38 months, which is considered as a reliable predictor

**Response: Thank you for the careful check. It has been revised.**

L70: Here it is worth mentioning that the QBO index at 20 hPa is used, different from studies for Northern Hemisphere.

**Response: Thank you for your comment. This sentence has been written as follows.**

*"Typically, when tropical stratosphere winds near 50 hPa are used to define the QBO phase, the NH stratospheric polar vortex during winter tends to strengthen during the WQBO and weaken during the EQBO (Baldwin et al. 2001; Anstey and Shepherd, 2014; Anstey et al., 2022). In the SH, the Antarctic stratospheric polar vortex shows responses to the tropical winds around 20 hPa (Baldwin and Dunkerton 1998; Baldwin et al., 2001; Baldwin and Dunkerton 1998; Anstey et al., 2022; Rao et al., 2023b). Therefore, this study uses the standardized zonal-mean zonal wind averaged over 10°S−10°N at 20 hPa to define the QBO phase."*

L74: Monthly => monthly

**Response: Thank you for the careful check. It has been revised.**

L95: mean stream function => mean mass stream function

**Response: Thank you for the careful check. It has been revised.**

L100: phi is geopotential or potential height? Please check.

**Response: Thank you for your comment. $\Psi$ is the geopotential height scaled by the Coriolis parameter, which is defined as follows (Harnik and Lindzen, 2001):**

$$\Psi = {}^{\phi}\!/\!_{f} \tag{R1}$$

**Reference**

**Harnik, N. and Lindzen, R. S.: The effect of reflecting surfaces on the vertical structure and variability of stratospheric planetary waves, J. Atmos. Sci., 58, 2872–2894, doi: 10.1175/1520-0469(2001)058<2872:TEORSO>2.0.CO;2, 2001.**

L107, 111: dm, what is the meaning of m?

**Response: Thank you for your comment. The $\int [s] dm$ is defined as $\int_z \int_y \int_x s\rho \, dx \, dy \, dz$ (Hu et al., 2004), and the $\rho$ is the air density. This point has been clarified in the revised manuscript.**
*"The $\int [s] dm$ is defined as $\int_z \int_y \int_x s\rho_0 \, dx \, dy \, dz$."*

**Reference**

**Hu, Q., Tawaye, Y. and Feng, S.: Variations of the Northern Hemisphere Atmospheric Energetics: 1948–2000, J. Climate, 17, 1975–1986, doi: 10.1175/1520-0442(2004)017<1975:VOTNHA>2.0.CO;2, 2004.**

L120: '' denotes meridional average? I did not see '' in all equations.

**Response: Thank you for your comment. The ″ represents departure from meridional average, and is used in calculation of the mean available potential energy (PM).**

$$PM = \frac{c_p}{2} \int \gamma [< T >]''^2 \, dm \tag{R2}$$

L125: 1.9*2.5 => 0.95*1.25

**Response: Thank you for your comment. We have examined the horizonal resolution of the model, which is 1.9° × 2.5°.**

L148: several months => Please tell the specific number.

**Response: Thank you for your comment. This sentence has been rewritten as:**

*"The QBO-SH polar vortex connection emerges in July and persists until austral late spring, with the effect peaking in November (i.e., the influence of the July QBO on the polar vortex peaks four months later, the influence of the August QBO peaks three months later, and so on). This result is consistent with the previous studies (Baldwin and Dunkerton, 1998; Yamashita et al., 2018)."*

L151: influence => influences

**Response: Thank you for the careful check. It has been revised.**

L158: This relation indeed exists, but the maximum response of the polar vortex to QBO is in austral spring.

**Response: Thank you for your comment. This sentence has been written as:**

*"The WQBO in winter (QBO index in July greater than 1) does not consistently lead to a stronger zonal-mean zonal wind in polar regions in spring, and the correction between them is only 0.23. These results suggest that the direct impact of the QBO on the Antarctic polar vortex is weak."*

L166: measure => depict

**Response: Thank you for the careful check. It has been revised.**

L168: and polar => and polar regions.

**Response: Thank you for the careful check. It has been revised.**

L239: negative anomalous temperature => negative temperature anomalies

**Response: Thank you for the careful check. It has been revised.**

L255: exhibit a … anomalies => remove a

**Response: Thank you for the careful check. It has been revised.**

L276: as a results => as a result

**Response: Thank you for the careful check. It has been revised.**

L356: moths => months?

**Response: Thank you for the careful check. It has been revised.**

L379: It is unknow which figure are based on the sensitivity experiments.

**Response: Thank you for your comment. Figure R6 (Figure 8 in the manuscript) shows the CESM model simulation results. The QBO nudging simulations with varying initial fields support our findings that the extratropical mode plays an important role in establishing the relationship between the WQBO and the Antarctic stratospheric polar vortex. In the revised manuscript, all figure captions have been verified, and data sources are now clearly specified.**

[Figure]

**Figure R6. (a) The zonal-mean zonal wind at 60ºS and 100 hPa in November is**

plotted against the extratropical mode's PC in July, derived from the 20 ensembles of CESM simulation. The boxplot describes a summary of these ensembles. The light grey box spans from the lower dectile to the upper dectile, and the dark grey spans the lower quartile to upper quartile. The lines inside the dark grey box marks the median zonal wind. The lower and upper whiskers indicate the minimum and maximum zonal wind among the 20 ensembles. Additionally, the blue line represents the linear fit between the extratropical mode and the median zonal wind, with their correlation coefficient displayed in the bottom right-hand corner. (b)−(f) Composite differences in zonal-mean zonal wind from July to November between the Pos-Exmode and Neg-Exmode according to CESM simulations. The dotted regions mark the differences between the Pos-Exmode and Neg-Exmode are statistically significant at the 95% confidence level.

L386: anomalous … wind => … wind anomalies

**Response: Thank you for the careful check. It has been revised.**

---

## Author Comment (AC3)

**Replies to Referee CC1**

**Manuscript number**: **EGUSPHERE-2024-2669**

**Title**: The Joint Effect of Mid-latitude Winds and the Westerly Quasi-Biennial Oscillation Phase on the Antarctic Stratospheric Polar Vortex and Ozone

**2024**

We thank the reviewer Dr. Yamashita and the editor for your helpful comments which helped us greatly to improve our paper. We modified our paper according to the comments. Our replies are summarized as below:

In the original manuscript, we classified the WQBO into WQBO-Strong Polar Vortex (W-SPV) and WQBO-Weak Polar Vortex (W-WPV) according to the phase of the extratropical mode in July (Figure R1a). However, we realized that the two phases are misnamed, as a positive extratropical mode does not always lead to a stronger polar vortex, despite the strong correlation between them. In the revised manuscript, we renamed the positive and negative extratropic mode in July as Positive-Extratropic mode (Pos-Exmode) and Negative-Extratropic mode (Neg-Exmode) as in Figure R1b.

[Figure]

Figure R1. The corresponding time series for the extratropic mode and polar mode of Figure 3 in the manuscript. (a) The Figure 3c in the original manuscript. (b) The Figure 3c in the revised manuscript.

This paper describes the relationship between the SH mid-latitudes wind and the Antarctic polar vortex under the westerly QBO condition. The relationship is derived with the statistical method. However, there is a lack of proper discussion about the dynamical linkage between them partially, as listed below. In addition, there is a lack

of proper discussion about the previous studies. Thus, I have a concern about the present manuscript.

**Response: Thank you for your comment. First, we should clarify that we have established a robust connection between the winter WQBO and the Antarctic stratospheric polar vortex in late austral spring through the statistical method (Figure 3c in the manuscript). Additionally, the underlying dynamical mechanisms have been analyzed (Figures R2−R4, Figures 5−7 in the manuscript), and CESM model simulations support our conclusions (Figures R5, Figure 8 in the manuscript).**

[revised manuscript text omitted]

p.2, L35-40: Baldwin and Dunkerton (1998) also mentioned "the zonal-mean zonal wind in the lower stratosphere decelerates more rapidly from September to October during the EQBO than during the WQBO".

**Response: Thank you for your comment. Baldwin and Dunkerton (1998) demonstrated that in the Southern Hemisphere (SH), the QBO modulates the vortex throughout the winter, with its largest influence occurring in late spring. However, their composite analysis indicated only a slightly cooling of the vortex during the westerly QBO phase (WQBO), and no statistical test to confirm these findings. In the original manuscript, we cited these articles to emphasize that the QBO-polar vortex coupling in the SH is not yet fully understood. We apologize if the original phrasing was unclear, and we have revised the sentence as follows.**

*"In the Southern Hemisphere (SH), upward-propagating planetary waves are weak due to the weaker thermal contrast between land and sea. Consequently, the QBO-vortex coupling, which is closely related to planetary waves, has received less attention than those in the Northern Hemisphere (Garcia and Solomon, 1987; Lait et al., 1989; Baldwin and Dunkerton, 1998; Naito, 2002; Hitchman and Huesmann, 2009; Yamashita et al. 2018; Rao et al., 2023a, 2023b).*"

**Reference**

**Baldwin, M. P. and Dunkerton, T. J.: Quasi-biennial modulation of the southern hemisphere stratospheric polar vortex, Geophys. Res. Lett., 25, 3343–3346, doi: 10.1029/98GL02445, 1998.**

p.2, L40, 45: The relationship between the below two sentences are not clear.

"the extratropical response to the QBO in late-winter SH can be interpreted as a modulation of the final warming by the QBO"

"the timing of the Antarctic stratospheric polar vortex's response to the QBO signal remains unclear"

**Response: Thank you for your comment. We agree that this part was not clearly written, and it has been revised as follows:**

*"Naito et al. (2002) examined the QBO signal in the SH and found that from September to October, the zonal-mean zonal wind in the lower stratosphere decelerates more rapidly during the EQBO than WQBO. This deceleration is attributed to the stronger upward wave propagation from the troposphere and larger wave convergence during the EQBO in these two months. However, Anstey et al. (2014) demonstrated that the extratropical response to the QBO occurs in November, interpreted as the modulation of the Antarctic polar vortex's final warming. Thus, it remains unclear when the response of Antarctic stratospheric polar vortex to the QBO reaches its peak."*

p.2, L45: "Wang et al. (2022) examine the QBO signals in stratospheric ozone." The implication of the Garcia and Solomon (1987) will be also mentioned in this context.

**Response: Thank you for your comment. This point has been added to the revised manuscript.**

*"In addition to the Antarctic stratospheric polar vortex, previous studies have also investigated the QBO signals in stratospheric ozone. Garcia and Solomon (1897) demonstrated that the year-to-year fluctuations in Antarctic ozone may be linked to the tropical QBO. In contrast, Wang et al. (2022) suggested that the Antarctic total column ozone (TCO) shows no significantly respond to the QBO signal (Figure 2 in Wang et al., 2022). So far, the impacts of the QBO on the Antarctic stratospheric polar vortex and ozone have not been well documented."*

**Reference**

**Garcia, R. R. and Solomon, S.: A possible relationship between interannual variability in Antarctic ozone and the quasi-biennial oscillation, Geophys. Res. Lett., 14, 848–851, doi: 10.1029/GL014i008p00848, 1987.**

p.3 L70: There is no reasonable explanation for why 20 hPa was chosen as a single pressure level.

**Response: Thank you for your comment. In the manuscript, we discussed the reason why a single-level wind (20 hPa) has been used to select the WQBO and EQBO years as follows:**

*"Previous studies have employed different methods to define the QBO index. Some are based on the tropical zonal-mean zonal wind at a single pressure level (Holton and Tan, 1980; Gray et al., 1992; Baldwin et al., 2001; Garfinkel and Hartmann, 2007), while others use two QBO indices on different pressure levels (Andrews et al., 2019) or use empirical orthogonal function (EOF) analysis applied on the tropical zonal-mean zonal wind (Randel et al., 1999; Anstey et al., 2010; Rao and Ren, 2018), which can better capture QBO's vertical structure. The QBO phase defined by the EOF method is similar to that defined by the single pressure level QBO index (Baldwin et al., 2001; Rao et al., 2020b). Typically, when tropical stratosphere winds near 50 hPa are used to define the QBO phase, the NH stratospheric polar vortex during winter tends to strengthen during the WQBO and weaken during the EQBO (Baldwin et al. 2001; Anstey and Shepherd, 2014; Anstey et al., 2022). In the SH, the Antarctic stratospheric polar vortex shows responses to the tropical winds around 20 hPa (Baldwin and Dunkerton 1998; Baldwin et al., 2001; Baldwin and Dunkerton 1998; Anstey et al., 2022; Rao et al., 2023b). Therefore, this study uses the standardized zonal-mean zonal wind averaged over 10°S−10°N at 20 hPa to define the QBO phase. The EQBO phase is defined as years when the tropical standardized zonal-mean zonal wind is less than −1, while the WQBO corresponds to years when the tropical standardized zonal-mean zonal wind is greater than 1."*

**To clarify further, the empirical orthogonal function (EOF) analysis is applied to**

the tropical zonal-mean zonal wind, averaged over 10°S–10°N from 10 hPa to 70 hPa (following Randel et al., 1999; Anstey et al., 2010; Rao and Ren, 2018). The first EOF mode depicts the in-phase changes in equatorial stratospheric zonal wind from the lower to the upper stratosphere (black line in Figure R6a), accounting for 56.04% of the total variance and showing the maximum correlation with the single-level equatorial wind at 20–30 hPa. The second EOF mode represents the contrasting variations between the lower and upper stratosphere (black line in Figure R6b), and it agrees well with the equatorial wind at 50 hPa (blue line in Figure R6b) variation. However, the second EOF mode explains only about 30% of the total variance, suggesting that the first EOF mode-like equatorial wind at 20–30 hPa serves as a good indicator of the QBO phase.

Additionally, years in WQBO are selected according to the equatorial wind at 20 and 30 hPa, respectively, as shown in Table R1. Note that the years identified by the 20 hPa zonal wind are largely contained by those at 30 hPa, except for 2022. Therefore, we use 20 hPa equatorial wind to define the QBO phase, consistent with previous studies (Baldwin et al., 2001; Naito, 2002; Rao et al., 2020).

[Figure]

Figure R6. We first obtain the first two principal components (PCs) of the monthly mean zonal-mean zonal wind, averaged over 10°S–10°N from 10 hPa to

70 hPa in July, through empirical orthogonal function (EOF) analysis. (a) The vertical structure of the in-phase QBO mode (black line), and the correlation between this PC and the tropical zonal-mean zonal wind at different levels (blue line). (b) Same as panel (a), but for the out-of-phase mode.

Table R1. Years categorized as WQBO based on the tropical zonal wind at 20 hPa and 30 hPa, respectively.

| Level | WQBO |
| --- | --- |
| 20 hPa | 1980, 1985, 1990, 1997, 2004, 2006, 2008, 2013, 2015, 2016, 2022 |
| 30 hPa | 1980, 1985, 1990, 1995, 1997, 1999, 2002, 2004, 2006, 2008, 2013, 2015, 2016, 2019 |

[Figure]

**Figure R7.** (a), (c), (e), (g), (i), and (k) Difference in temperature between the CESM model simulation and JRA55 reanalysis dataset at 50 hPa on 1 September 2021 from different ensembles. (b), (d), (f), (h), (j), and (l) Difference in zonal wind between the CESM model simulation and JRA55 reanalysis dataset on 1 September 2021.

[Figure]

Figure R8. (a) The climatological mean geopotential height, averaged over 20°S–20°N and derived from JRA55 reanalysis dataset from 1980 to 2022, is shown as the thick black line. This geopotential height averaged over 20°S–20°N is decomposed into wave numbers 1, 2, and 3 using the simple Fourier analysis, with their respective amplitudes represented by the thin grey, blue, and red lines, respectively. The X-axis represents the geopotential height in logarithmic coordination. (b) The climatological mean geopotential height deviations from the zonal mean at 50 hPa, derived from the JRA55 reanalysis dataset from 1980 to 2022. The dashed lines indicate the full nudging regions.

**Response: Thank you for your comment. We agree that discrepancies may exist between different reanalysis dataset, and validating the results across multiple datasets can enhance their robustness. First, we present the WQBO years selected based on the MERRA-2 and JRA55 reanalysis datasets, respectively, in Table R2. The WQBO years show strong agreement between these two datasets. Since only 20 hPa equatorial winds are used to identify the QBO phase, the absence of 40 hPa data in JRA55 does not affect our results.**

**Secondly, we reprinted Figure 3 (shown in Figure R9 here) of the manuscript using the JRA55 reanalysis dataset (Figure R10). It also shows a strong correlation between the extratropic mode in July and the Antarctic polar vortex in late austral spring, suggesting that the conclusions in the original manuscript are not influence by the choice of reanalysis dataset. Additionally, in the manuscript, we clarified the use of different reanalysis datasets, and the consistency of the results across both datasets further demonstrates the robustness of our findings as follows:**

*"Note that different sets of reanalysis data are used to force the CESM and perform other analysis. The consistency of the results across both datasets indicates the robustness of the findings."*

[Figure]

**Figure R9 Spatial patterns for the first paired mode of the (a) monthly mean zonal-mean zonal wind over 0−40°S and 1−70 hPa in July, i.e., extratropical mode (b) zonal-mean zonal wind over 50°S−70°S and 1−70 hPa in November by the singular value decomposition (SVD) analysis during WQBO years, based on the MERRA-2 reanalysis dataset from 1980 to 2022. The right panel of (a) is the profile of the extratropical mode averaged over 0−5°S. The variance explained by the first mode is shown in the top right-hand corner. (c) The corresponding time series for the paired mode, with their correlation coefficient are shown in the bottom right-hand corner (text in blue). The solid blue line represents the linear regression of the extratropical mode and polar mode. The size and color of the circle markers in panel (c) are proportional to the Niño 3.4 index, with yellow dots indicating a positive Niño 3.4 index and blue dots indicating a negative Niño 3.4 index. The standardized ozone volume mixing ratios, averaged over 60°S−90°S at 70 hPa in November, against the extratropical mode time series are shown in triangular markers (right Y-axis), with their correlation coefficient displayed in the top left-hand corner (text in black). The dashed black line represents the linear regression of the extratropical mode and ozone in November.**

[Figure]

**Figure R10. Same as Figure R9, but it derived from JRA55 reanalysis datasets.**

**Table R2. Years categorized as WQBO based on the MERRA-2 and JRA55 reanalysis datasets, respectively.**

| Dataset | WQBO |
|---------|------|
| MERRA-2 | 1980, 1985, 1990, 1997, 2004, 2006, 2008, 2013, 2015, 2016, 2022 |
| JRA55 | 1980, 1985, 1990, 1997, 2004, 2006, 2008, 2013, 2015, 2016, 2022 |

p.6, L145: "consistently reaching their maximum in November": This result will be compared to that of Baldwin and Dunkerton (1998).

**Response: Thank you for your comment. This sentence has been written as follows.**

*"The QBO-SH polar vortex connection emerges in July and persists until austral late spring, with the effect peaking in November (i.e., the influence of the July QBO on the polar vortex peaks four months later, the influence of the August QBO peaks three months later, and so on). This result is consistent with the previous studies (Baldwin and Dunkerton, 1998; Yamashita et al., 2018)."*

p.6, L155: The "direct impact" is not clear. The results only show that the correlation between the QBO index and the zonal wind at 60S, 70hPa.

**Response: Thank you for your comment. In the manuscript, we first analyzed**

the correlation between the winter QBO index and the spring Antarctic polar vortex (Figure 1b in the manuscript). However, the correlation is only 0.23, indicating that the winter WQBO (EQBO) does not consistently lead to a stronger (weaker) Antarctic polar vortex in late austral spring, as previous suggested (Baldwin et al., 2001; Naito, 2002; Anstey et al., 2010). We described this as the 'direct impact' of the QBO on the Antarctic polar vortex. We further proposed that when the QBO aligns with mid-latitudes wind in the upper stratosphere, the predictability of the spring Antarctic polar vortex's strength could be enhanced (Figure R9c).

Despite the correlation between the QBO index and the zonal wind at 60°S and 70 hPa, Figure R11 shows the composited difference in zonal-mean zonal wind between the WQBO and EQBO. Even with a large sample size of 22 in this composite analysis, it remains difficult to achieve statistical significance, indicating high sample variance. This further suggests that the winter WQBO (EQBO) cannot consistently lead to a stronger (weaker) Antarctic polar vortex in late austral spring.

[Figure]

Figure R11. Composite differences in zonal-mean zonal wind anomalies in November between the WQBO and EQBO according to MERRA-2 reanalysis

dataset from 1980 to 2022. The phase of QBO is defined by using equatorial wind data in July.

Response: Thank you for your comment. In the manuscript, we refer to the 'extratropical zonal-mean zonal wind in the upper stratosphere' as the 'extratropical mode' (Figure R9a). It is not formed through the secondary circulation of the QBO. If influenced by the QBO phase, the winter extratropical mode would exhibit opposite phases between the WQBO and EQBO. Figure R12c shows the composited difference in zonal-mean zonal wind in austral winter (July) between the WQBO and EQBO. No uniform positive zonal-mean zonal wind anomalies are observed between 20°S and 40°S at 1−50 hPa, suggesting that the 'winter extratropical mode' discussed in the manuscript is not influenced by the QBO phase. This is an interesting question that warrants further investigation, and we plan to explore this in future work.

[Figure]

**Figure R12. (a) Composited zonal-mean zonal wind in July during the WQBO phase. (b) Same as panel (a), but for the EQBO phase. (c) Composited difference in zonal-mean zonal wind between the WQBO and EQBO. The dotted regions mark the differences are statistically significant at the 95% confidence level.**

p.6, L180: "Thus, we can conclude that the winter extratropical mode, in conjunction with the WQBO, is closely linked to the spring Antarctic polar vortex and ozone.": This probably responds of the upper stratospheric vortex shift to the upper in July, together with the poleward and downward movement from winter to summer (Yamashita et al. 2018, DOI:10.2151/jmsj.2018-057).

**Response: Thank you for your comment. First, we agree that the center of the stratospheric polar vortex shifts poleward and downward from winter to spring, reflecting its seasonal variation (dark green lines in the first and second columns of Figure R13). Compared with the Neg-Exmode events, the center of Antarctic polar vortex during Pos-Exmode is located at lower latitudes, resulting in the dipole pattern in Figure R13k. This dipole pattern evolves and shifts poleward and downward, ultimately leading to a strong polar vortex in November through wave-mean flow interactions (Figures R13k–o). Therefore, in the manuscript, we conclude that this winter extratropical mode under the WQBO is closely linked to the spring Antarctic polar vortex and ozone.**

[Figure]

**Figure R13. Monthly mean zonal-mean zonal wind from July to November. (a)–(e) Zonal-mean zonal wind averaged over Pos-Exmode years derived from MERRA-2 reanalysis dataset (the first column). The contours indicate the climatological zonal-mean zonal wind from 1980 to 2022, and the contour intervals are 10 m/s. (f)–(j) Zonal-mean zonal wind averaged over Neg-Exmode years (the second column). (k)–(o) Composite difference in zonal-mean zonal wind between the Pos-Exmode and Neg-Exmode years. The dotted regions mark the differences are statistically significant at the 95% confidence level.**

Secondly, while this dipole mode in July seems similar to that in Figure 4a (Figure R14) of Yamashita et al. (2018), the underlying mechanisms are entirely different. Since our analysis of Pos-Exmode and Neg-Exmode events are both under the WQBO phase, it emphasizes that during WQBO, the anomalous westerly winds in the mid-latitudes of the upper stratosphere are essential for strengthening the polar vortex in austral spring.

[Figure]

**Figure R14. QBOa coefficients of zonal wind in June, July, August, September,**

and October, together with climatology (cite from Yamashita et al., 2018).

$$\frac{d\overline{u}}{dt} - f_0 \overline{v}^* - \overline{X} = \rho_0^{-1} \nabla F \tag{R1}$$

**Where the convergence of planetary waves (negative $\nabla F$ ) is balanced by the decelerated jet stream (negative $\frac{d\overline{u}}{dt}$ ) and strengthen Brewer-Dobson (B-D) circulation (positive $f_0 \overline{v}^*$ ).**

**In the manuscript, we concluded that the anomalous E-P flux divergence, along with its poleward and downward shift, results in the positive anomaly in zonal-mean zonal wind and a stronger polar vortex in November. We attributed**

these anomalous E-P flux divergences to anomalous positive stream function (Figure R15a). A question arises to whether the anomalous positive stream function is the result of the anomalous E-P flux divergence (Figure R15b). The causality should be clarified here.

First, we replotted the Figure 7 of the original manuscript, ensuring that the $\frac{d\overline{u}}{dt}$ and $\nabla F$ share the same unit for consistency (Figure R16). From July to August, positive $-f_0\overline{v}^*$ anomalies are located equatorward of the E-P flux divergence and are much larger in magnitude (Figures R16b and l), suggesting that the E-P flux divergence cannot fully account for the positive stream function anomaly. In contrast, from July to November, the location and magnitude of positive $\frac{d\overline{u}}{dt}$ anomalies align well with the positive anomalous E-P flux divergence. We agree that while the positive anomalous E-P flux divergence might contribute to the positive $-f_0\overline{v}^*$ anomalies, its contribution is relatively minor. Thus, the mechanisms depicted in Figure R15a is in accord with the present findings.

[Figure]

Figure R15. (a) The mechanisms of how the positive winter extratropical mode induce a stronger Antarctic polar vortex in November under the WQBO phase in

**the manuscript. (b) The reviewer's comment on our mechanisms.**

[Figure]

**Figure R16. Composite differences in the (a)−(e) the divergence of E-P flux anomalies, (f)−(j)** $\dfrac{d\bar{u}}{dt}$ **anomalous, and (k)−(o)** $-f\bar{v}^*$ **from July to November between the Pos-Exmode and Neg-Exmode according to MERRA-2 reanalysis dataset. The dotted regions mark the differences in (a)−(e) the divergence of E-P flux, (f)−(j)** $\dfrac{d\bar{u}}{dt}$ **, and (k)−(o)** $-f\bar{v}^*$ **between the Pos-Exmode and Neg-Exmode are statistically significant at the 90% confidence level.**

Secondly, the time series of the difference in E-P flux divergence (purple line) and $\overline{w}^*$ (yellow line) between Pos-Exmode and Neg-Exmode events are shown in Figure R17. Note that the peak value of $\overline{w}^*$ (orange dashed lines) precedes the peak of E-P flux divergence (purple dashed lines), suggesting that the anomalous upward motion drives the positive E-P flux divergence anomalies. This lead-lag relationship between the $\overline{w}^*$ and E-P flux divergence further support the mechanisms depicted in Figure R15a, where positive stream function anomalies result in E-P flux divergence. Additionally, the model simulation also supports this point (Figure R18).

[Figure]

Figure R17. Composite evolutions of the difference in E-P flux divergence (purple line) and $\overline{w}^*$ (yellow line) between Pos-Exmode and Neg-Exmode events according to the MERRA-2 reanalysis dataset. Thick lines indicate that the differences are significant at the 90% confidence level according to the Student's *t*-test. The dashed lines indicate the times when E-P flux divergence and $\overline{w}^*$ reach their peak values.

[Figure]

**Figure R18. Same as Figure R17, but the data is derived from the CESM simulations.**

Additionally, the sensitivity nudging simulations provide additional confirmation of these findings. First, we calculated the composite meteorological fields for WQBO, Pos-Exmode and Neg-Exmode. Nudging is then applied to the WQBO stratospheric zonal wind, meridional wind, and temperature fields from 1 to 100 hPa between 20°S and 20°N, using a nudging coefficient of 1.0 to simulate WQBO forcing. Secondly, the meteorological fields from 1 to 70 hPa and 20°S to 40°S are nudged to the Pos-Exmode and Neg-Exmode fields, respectively, to simulate the extratropical forcing described in the manuscript. Then the Whole Atmosphere Community Climate Model (WACCM) in the Community Earth System Model version 2 (CESM2) to conduct 7 ensemble experiments. The difference between the Pos-Exmode and Neg-Exmode simulations represents the atmospheric response to the extratropical mode during WQBO phase.

Figure R19a shows the composite difference in zonal-mean zonal wind between the Pos-Exmode and Neg-Exmode simulations in July, derived from CESM2 simulations. Note positive and negative zonal-mean zonal wind anomalies are observed at 30°S and 60°S between 1 hPa and 70 hPa, respectively, which

resembles those in Figure 4a of the original manuscript. This forcing can lead to a consistent positive stream function anomaly across the 7 ensembles. However, the E-P flux divergence anomalies do not exhibit a consistent positive value, suggesting that the extratropical mode align with the WQBO can induce a positive stream function (Figure R15a) rather than the E-P flux divergence (Figure R15b). Although these sensitive simulations cannot consistently reproduce positive E-P flux divergence anomalies due to model nudging for only one months and the discrepancy among the initial fields, the downward and poleward shift of the anomalous zonal-mean zonal wind, E-P flux divergence, and the stream function can be reproduced (Figure R20).

[Figure]

**Figure R19. (a) Composite difference in zonal-mean zonal wind between the Pos-Exmode and Neg-Exmode simulations in July derived from CESM2 sensitive simulations. The dotted regions mark the differences between the Pos-Exmode and Neg-Exmode simulations are statistically significant at the 90% confidence level. (b) Composite difference in stream function averaged over 1−30 hPa and 0°−40°S between the Pos-Exmode and Neg-Exmode simulations in July derived from CESM2 simulations. (c) Same as panel (b), but for the E-P flux divergence**

averaged over 1−30 hPa and 40°−70°S.

[Figure]

**Figure R20. (a)−(e) Composite difference in zonal-mean zonal wind between the Pos-Exmode and Neg-Exmode simulations from July to November derived from CESM2 sensitive simulations. The dotted regions mark the differences between the Pos-Exmode and Neg-Exmode simulations are statistically significant at the 90% confidence level. (f)−(j) Same as panels (a)−(e), but for the E-P flux divergence. (k)−(o) Same as panels (a)−(e), but for the stream function.**

Response: Thank you for your comment. Here, we use TOMCAT/SLIMCAT global three-dimensional offline chemical transport model (CTM) to estimate the Polar stratospheric cloud (PSC) areas. This model can capture the seasonal and interannual verbalities of PSCs well, as evaluated in our other study (Li et al., 2024). Additionally, the CTM can estimate the chemical ozone loss rate driven by chlorine catalytic reactions. Figure R21 compares the PSC area and ozone loss rate from chlorine catalytic reactions between Pos-Exmode and Neg-Exmode events. During Pos-Exmode, lower temperature in the Antarctic lower stratosphere from September to October lead to larger PSC areas than in Neg-Exmode events, resulting in more chemical ozone loss (absolute value of the ozone loss rate). Therefore, we concluded that "*Lower temperature in the lower stratosphere favors increased PSC area and chemical ozone depletion in the polar regions (Figs.4m−n). Consequently, the temperature decrease induced by the positive extratropical mode leads to negative anomalous ozone in the Antarctic lower stratosphere from September to November (Figs. 4m−o).*" The difference in PSC area between the Pos-Exmode and Neg-Exmode has been added in the revised manuscript (as shown in Figure R22 here).

[Figure]

**Figure R21. (a) Composited PSC area at 475 K isentropic level from September to November during the Pos-Exmode and Neg-Exmode years derived from SLIMCAT simulations. (b) Same as panel (a), but for the composited ozone loss rate averaged over 63°S−90°S.**

[Figure]

**Figure R22. Composite differences in (a)−(e) the zonal-mean zonal wind anomalies, (f)−(j) the scaled E-P flux anomalies (green vectors; horizontal component unit: 107 kg s−2; vertical component unit: 105 kg s−2) and the E-P flux divergence anomalies (shadings), (k)−(o) the zonal-mean temperature anomalies (shadings) and the zonal-mean ozone volume mixing ratio anomalies (contours; dashed lines are negative, and thick lines are zero contours. The contour intervals are 200 ppbv) from July to November between the Pos-Exmode and Neg-Exmode according to MERRA-2 reanalysis dataset. Composite differences in PSC areas between the Pos-Exmode and Neg-Exmode, based on the NASA ozone watch dataset from September to October, are shown at the top of Panels (m) and (n). The E-P flux and its divergence are calculated from wave 1 to 3. E-P flux vectors are scaled by the factor cosφ and multiplied by the square**

root of 1000.0/p in both the vertical and horizontal directions, where p is pressure in hPa. The dotted regions mark the differences in zonal-mean zonal wind, E-P flux divergence, and zonal-mean temperature are statistically significant at the 90% confidence level. Green shading marks the regions where the differences in ozone volume mixing ratio between the Pos-Exmode and Neg-Exmode are statistically significant at the 90% confidence level. Only the significant (at the 90% confidence level) E-P flux vectors have been plotted in panels (f)−(j).

**Response: Thank you for your comment. North of 30°S above 10 hPa, a negative vertical gradient of zonal-mean zonal wind anomaly induces a negative stream function anomaly, accompanied by anomalous downward motion (dashed lines in Figure R23a). The mechanism is similar to that in Figure 5 of the manuscript. In this case, the negative zonal-mean zonal wind shear anomaly is balanced by a colder center to the north and a warmer center to the south. These temperature anomalies are sustained by the anticlockwise secondary vertical motion. Additionally, in Figure R23, the location of positive temperature anomalies aligns well with the downward motion, indicating adiabatic heating associated with these downward motions.**

[Figure]

**Figure R23. Composite differences in the temperature anomalies (shadings) and vertical component of residual mean circulation ($\overline{w}*$) anomalies (contours; dashed lines are negative, and thick lines are zero contours. The contour intervals are 0.1 mm/s). The dotted regions mark the differences in temperature between the Pos-Exmode and Neg-Exmode are statistically significant at the 90% confidence level. Green shading marks the regions where the differences in $\overline{w}*$ between the Pos-Exmode and Neg-Exmode are statistically significant at the 90% confidence level.**

p.12, L265: "induced by the extratropical mode": I suppose that the B-D circulation difference is induced by the difference in the E-P flux divergence in the TEM terms.

**Response: Thank you for your comment. We acknowledge that the anomalous E-P flux divergence can lead to a weak B-D circulation and, to some extent, a positive anomalous stream function. However, in Figure R16, the positive $-f_0 \overline{v}^*$ anomalies are located equatorward of the E-P flux divergence and are much larger in magnitude, suggesting that the E-P flux divergence alone cannot fully account for the secondary circulation anomaly from July to August. Therefore, we agree that while the positive anomalous E-P flux divergence might contribute to the secondary circulations, its contribution is relatively minor in our study.**

**Secondly, in Figures R17 and R18, the peak value of $\overline{w}*$ (orange dashed lines) precedes the peak of E-P flux divergence (purple dashed lines), suggesting that**

**the anomalous upward motion drives the positive E-P flux divergence anomalies. This lead-lag relationship between the $\overline{w}*$ and E-P flux divergence further support the mechanisms depicted in Figure R15a, where positive stream function anomalies result in E-P flux divergence.**

**Additionally, the sensitive test from CESM2 nudging simulations also support these findings. In Figure R19, extratropical forcing consistently leads to a positive stream function anomaly across the 7 ensembles. However, the E-P flux divergence anomalies do not exhibit a consistent positive value, suggesting that the extratropical mode align with the WQBO can induce secondary circulation anomalies rather than the E-P flux divergence (Figure R15).**

p.13, L275-280: "which dominates the wave refractive index and results in anomalous E-P flux divergence around 50S in the upper stratosphere in July": Please indicate the contribution of the PV anomaly to the total wave refractive index.

**Response: Thank you for your comment. Here, we calculated the quasi-geostrophic refractive index (RI; Chen and Robinson, 1992), which is shown in equation R2.**

$$\mathbf{RI} = \underbrace{\frac{\overline{q_\varphi}}{\overline{u}}}_{term\,1} \underbrace{- \left( \frac{k}{a\cos\varphi} \right)^2 - \left( \frac{f}{2NH} \right)^2}_{term\,2} \tag{R2}$$

**Where the zonal mean potential vorticity meridional gradient $\overline{q_\varphi}$ is defined as follows:**

$$\overline{q_\varphi} = \frac{2\Omega}{a}\cos\varphi - \frac{1}{a^2}\left[ \frac{\left(\overline{u}\cos\varphi\right)_\varphi}{a\cos\varphi} \right]_\varphi - \frac{f^2}{\rho_0}\left( \rho_0 \frac{\overline{u_z}}{N^2} \right)_z \tag{R3}$$

**The $H$, $k$, $N^2$, $a$, and $\Omega$ are the scale height, zonal wave number, buoyancy frequency, Earth's radius and angular frequency, respectively. The $N^2$ is defined**

as:

$$N^2 = \frac{R}{H}\left(T_z + \frac{\kappa T}{H}\right)$$  (R4)

Where the $R$ is the specific gas constant for dry air, and the $\kappa = R/c_p$.

The RI consists of the zonal mean potential vorticity meridional gradient term and two additional terms. Figure R24 presents the climatological mean RI along with its components. Note that except south of 85°S, the term1 accounts for nearly all the variation in RI. In the region between 60°S and 70°S above 5 hPa (red rectangle), where RI is essential to the E-P flux divergence in Figure 4f of the manuscript, term2 appears larger than term1 in the climatological mean. However, we have confirmed that the sign of RI (Figure 25a) in this region is primarily determined by the negative anomalies in the potential vorticity meridional gradient term (term1 in R2, Figure R25b). The conditions in the subsequent four months are similar to those in July, so we do not describe them again in detail.

[Figure]

Figure R24. (a) Climatological mean RI, averaged in July from 1980 to 2022, derived from the MERRA2 reanalysis dataset. (b) Zonal mean potential vorticity meridional gradient term (the first term on the right side of R2). (c) The term2 on the right side of R2.

[Figure]

**Figure R25.** Composited difference in (a) RI calculated by R1, (b) term1, and (c) term2 between the Pos-Exmode and Neg-Exmode in July derived from MERRA-2 reanalysis dataset.

**Reference**

**Chen, P., and Robinson, W. A.: Propagation of planetary waves between the troposphere and stratosphere, J. Atmos. Sci., 49, 2533–2545, doi: 10.1175/15200469(1992)049<533:POPWBT>2.0.CO;2, 1992.**

Figure 7: The unit of the du/dt must be the same as that of E-P flux divergence. In addition, the fv* term will be also discussed in this context.

**Response: Thank you for your comment. Figure 7 of the original manuscript has been replotted as Figure R26, converting the unit of E-P flux divergence to m s$^{-1}$ day$^{-1}$ to match the unit of du/dt. Same as those in the original manuscript, the anomalous E-P flux divergence induces the poleward and downward shift of the positive anomalous zonal-mean zonal wind.**

**Additionally, the $-f_0\overline{v}^*$ align with E-P flux divergence and $\dfrac{d\overline{u}}{dt}$ are show in Figure R16. First, from July to August, positive $-f_0\overline{v}^*$ anomalies are located equatorward of the E-P flux divergence and are much larger in magnitude,**

suggesting that the E-P flux divergence cannot fully account for the positive stream function anomaly. In contrast, from July to November, the location and magnitude of positive $\dfrac{d\bar{u}}{dt}$ anomalies align well with the positive anomalous E-P flux divergence. We agree that while the positive anomalous E-P flux divergence might contribute to the positive $-f_0\bar{v}^*$ anomalies, its contribution is relatively minor.

[Figure]

Figure R26. Composite differences in the $\dfrac{d\bar{u}}{dt}$ anomalies (color shadings) and E-P flux divergence anomalies (contours; dashed lines are negative, and thick

lines are zero contours. The contour intervals are 0.25 m s$^{-1}$ day$^{-1}$) from July to November between the Pos-Exmode and Neg-Exmode according to MERRA-2 reanalysis dataset. The dotted regions and green shadings mark the differences in $\dfrac{d\bar{u}}{dt}$ and the differences in E-P flux divergence between the Pos-Exmode and Neg-Exmode are statistically significant at the 90% confidence level.

p.18, L385: "pushing the positive anomalous zonal-mean zonal wind towards the pole": The causality is not derived from this analysis, as mentioned above.

**Response: Thank you for your comment. Figures R15 to R20 provide additional validation of the causality. In summary, the positive extratropical mode under the WQBO initiates a secondary circulation, which subsequently modifies the PV distribution and planetary wave propagation, ultimately strengthening the polar vortex in November.**

p.18, L395: "we propose an upper stratospheric pathway for the QBO's impact on the stratospheric polar vortex": The comparison to the Yamashita et al. (2018)'s results will be needed in this context.

**Response: Thank you for your comment. We have compared with the Yamashita et al. (2018) in the introduction in the revised manuscript.**

*"Yamashita et al. (2018) examined the influence of the QBO on SH extratropical circulation from austral winter to early summer using a multiple linear regression approach. Their findings suggest that the QBO-SH polar vortex connection operates through two distinct pathways: the mid-stratospheric pathway, which tends to suppress the propagation of planetary waves into the stratosphere during WQBO, and the low-stratospheric pathway, which tends to enhance upward planetary waves in EQBO. The QBO-SH polar vortex connections established by Yamashita et al. (2018) are statistically significant."*

**Reference**

**Yamashita, Y., Naoe, H., Inoue, M. and Takahashi, M.: Response of the Southern Hemisphere Atmosphere to the Stratospheric Equatorial Quasi-Biennial Oscillation (QBO) from Winter to Early Summer, J. Meteorol. Soc. Jpn., 96, 6, 587−600, doi: 10.2151/jmsj.2018-057, 2018.**

Other comments

Figure 2(c): "Sepetmber" -> "September"

**Response: Thank you for your comment. It has been corrected**

Figure 3: The "solid blue line" is not found.

**Response: Thank you for your comment. Figure 3c from the original manuscript is now presented as Figure R27. The solid blue line represents the linear fit between the extratropical mode and polar mode, with the correlation coefficient displayed in the lower-right corner of the panel.**

[Figure]

**Figure R27. The corresponding time series for the paired mode, with their correlation coefficient shown in the bottom right-hand corner (text in blue). The solid blue line represents the linear regression of the extratropical mode and polar mode. The size and color of the circle markers are proportional to the Niño 3.4 index, with yellow dots indicating a positive Niño 3.4 index and blue dots indicating a negative Niño 3.4 index. The standardized ozone mixing ratios, averaged over 60°S−90°S at 70 hPa in November, against the extratropical mode time series are shown with triangular marker (right Y-axis), with their**

correlation coefficient displayed in the top left-hand corner (text in black). The dashed black line represents the linear regression of the extratropical mode and ozone in November.